# Activation of goblet-cell stress sensor IRE1β is controlled by the mucin chaperone AGR2

Eva Cloots [1,2], Phaedra Guilbert [1,2], Mathias Provost [3,4], Lisa Neidhardt [5], Evelien Van de Velde[1,2], Farzaneh Fayazpour [1,2], Delphine De Sutter[6,7], Savvas N Savvides[3,4], Sven Eyckerman[6,7,8] & Sophie Janssens [1,2,8 ✉]

## Abstract

**Intestinal goblet cells are secretory cells specialized in the production of mucins, and as such are challenged by the need for efficient protein folding. Goblet cells express Inositol-Requiring Enzyme-1β (IRE1β), a unique sensor in the unfolded protein response (UPR), which is part of an adaptive mechanism that regulates the demands of mucin production and secretion. However, how IRE1β activity is tuned to mucus folding load remains unknown. We identified the disulfide isomerase and mucin chaperone AGR2 as a goblet cell-specific protein that crucially regulates IRE1β-, but not IRE1α-mediated signaling. AGR2 binding to IRE1β disrupts IRE1β oligomerization, thereby blocking its downstream endonuclease activity. Depletion of endogenous AGR2 from goblet cells induces spontaneous IRE1β activation, suggesting that alterations in AGR2 availability in the endoplasmic reticulum set the threshold for IRE1β activation. We found that AGR2 mutants lacking their catalytic cysteine, or displaying the disease-associated mutation H117Y, were no longer able to dampen IRE1β activity. Collectively, these results demonstrate that AGR2 is a central chaperone regulating the goblet cell UPR by acting as a rheostat of IRE1β endonuclease activity.**

**Keywords** AGR2; Goblet Cells; IRE1β; Mucus Homeostasis; Unfolded Protein Response (UPR)
**Subject Categories** Digestive System; Post-translational Modifications & Proteolysis; Translation & Protein Quality

See also: L Neidhardt et al (2023)

## Introduction

The gastrointestinal (GI) tract is protected by an essential mucus layer, which forms the main physical barrier between the intestinal epithelium and the outside world. The main protein component of the intestinal mucus layer is the glycoprotein MUC2, which is produced by goblet cells, specialized secretory cells that are present both in the small and large intestine (Birchenough et al, 2015). MUC2 is a large glycoprotein of over 5100 amino acids that becomes assembled in disulfide bond stabilized dimers in the endoplasmic reticulum (ER) before translocating to the Golgi (Birchenough et al, 2015; Fass and Thornton, 2023; Sharpe et al, 2018). As such, its folding poses a significant burden on the ER, making goblet cells exquisitely sensitive to folding defects (Heazlewood et al, 2008). This is further supported by the close link between aberrations in mucus folding capacity and inflammatory bowel disease (Bergstrom et al, 2017; Heazlewood et al, 2008; Johansson et al, 2014).

To ensure proper folding, eukaryotes developed a sophisticated adaptive response termed the unfolded protein response (UPR) orchestrated by three ER-based sensors that respond to changes in folding load: PKR-like ER Kinase (PERK), Activating Transcription factor (ATF)6 and inositol-requiring enzyme-1 (IRE1) (gene name *ERN1*) (Hetz et al, 2020). Together, they coordinate a program to slow translation, expand the ER, upregulate the expression of chaperones, and amplify the capacity to process misfolded proteins for degradation. IRE1 is the most evolutionarily conserved member of the three UPR sensors and is characterized by an N-terminal ER-luminal sensor domain, a Type I transmembrane domain, and two C-terminal cytoplasmic enzymatic domains: a kinase and an endonuclease domain (Cloots et al, 2021b). In steady-state conditions, IRE1 is kept in an inactive monomeric state by binding the chaperone-binding immunoglobulin Protein (BiP) (Bertolotti et al, 2000; Preissler and Ron, 2019). Upon accumulation of unfolded proteins in the ER, BiP dissociates from IRE1α, leading to its default dimerization/oligomerization and transphosphorylation (Ali et al, 2011; Amin-Wetzel et al, 2019; Bertolotti et al, 2000). Nucleotide-binding stabilizes the back-to-back dimer interface necessary for endonuclease activity (Korennykh et al, 2011). While

[1]Laboratory for ER stress and Inflammation, VIB Center for Inflammation Research, 9052 Ghent, Belgium. [2]Department of Pediatrics and Internal Medicine, Ghent University, 9052 Ghent, Belgium. [3]Unit for Structural Biology, VIB Center for Inflammation Research, 9052 Ghent, Belgium. [4]Unit for Structural Biology, Department of Biochemistry and Microbiology, 9052 Ghent, Belgium. [5]Cambridge Institute for Medical Research, University of Cambridge, Cambridge CB2 0XY, UK. [6]VIB Center for Medical Biotechnology, 9052 Ghent, Belgium. [7]Department of Biomolecular Medicine, Ghent University, 9052 Ghent, Belgium. [8]These authors contributed equally: Sven Eyckerman, Sophie Janssens. ✉E-mail: Sophie.Janssens@irc.vib-ugent.be

the details have not been fully elucidated yet, binding of unfolded proteins to the luminal domain has been postulated to induce IRE1 oligomerization (Credle et al, 2005; Karagöz et al, 2017). This creates a composite RNA binding pocket that accommodates IRE1's main endonuclease substrate, X-box binding protein 1 (*Xbp1*) mRNA, allowing excision of a 26 nt intron. The resulting frameshift leads to the production of the active transcription factor XBP1s (Yoshida et al, 2001). Apart from *Xbp1*, IRE1α has also been shown to target additional mRNAs for degradation in a process termed Regulated IRE1-Dependent Decay or RIDD, which further relieves the folding burden on the ER (Hollien et al, 2009; Hollien and Weissman, 2006).

While IRE1α is ubiquitously expressed, goblet cells and other epithelial cells in the GI and respiratory tract, uniquely express a second paralogue of IRE1, IRE1β (gene name *ERN2*) (Bertolotti et al, 2001; Cloots et al, 2021a; Cloots et al, 2021b; Wang et al, 1998), with expression levels that largely exceed those of IRE1α (Grey et al, 2020). IRE1β arose through whole genome duplication in ancestral vertebrates and analysis of IRE1β sequence variations in different vertebrate species suggested that IRE1β adopted a neofunctionalization with the emergence of a mucus-based system of barrier immunity (Cloots et al, 2021b). This, together with its unique expression in epithelial cells lining mucosal interfaces suggests that IRE1β functions in mucosal homeostasis, which is supported by earlier studies. IRE1β$^{-/-}$ mice are more susceptible to dextran sulfate sodium (DSS)-induced colitis, exhibiting worse inflammation and premature lethality (Bertolotti et al, 2001). The absence of IRE1β leads to a defect in goblet cell numbers (Tschurtschenthaler et al, 2017) and a recent study assigned a role for IRE1β in goblet cell maturation, mucin secretion and mucus barrier assembly, that was strictly dependent on the present of an intact microbiome (Grey et al, 2022). At a mechanistic level, Tsuru et al showed that IRE1β is responsible for trimming *Muc2* mRNA pools in a RIDD dependent manner, thereby avoiding overloading of the ER with MUC2 polypeptide chains (Tsuru et al, 2013). IRE1α and IRE1β show divergent roles in intestinal homeostasis. While loss of IRE1β affects goblet cell number, this has not been observed in IRE1α deficient mice (Tschurtschenthaler et al, 2017; Tsuru et al, 2013). Furthermore, IRE1β appears to protect from Crohn's disease like ileitis, triggered upon absence of XBP1 and ATG16L1 in Paneth cells, while IRE1α contributes to the disease (Tschurtschenthaler et al, 2017).

How IRE1β exerts these goblet cell-specific functions remains largely enigmatic. Despite its close homology to IRE1α, IRE1β displays only weak endonuclease activity, and even acts as a negative regulator of IRE1α endonuclease activity, both in vitro (Grey et al, 2020) and in vivo (Tschurtschenthaler et al, 2017). IRE1β only marginally responds to classical ER triggers and predominantly forms dimers instead of oligomers, consistent with a presumed prominent role for IRE1β in RIDD rather than in XBP1 splicing (Grey et al, 2020; Nakamura et al, 2011; Tam et al, 2014). One of the most remarkable properties of IRE1β—and explaining at least partially its poorly understood function—is the fact that overexpression of the protein in cell culture leads to a hyperactive and deleterious state. HeLa cells undergo rapid cell death upon IRE1β expression, suggested to be caused by 28S rRNA degradation (Iwawaki et al, 2001). At present, it is incompletely understood how IRE1β is regulated and how the difference in activity levels in distinct model systems can be explained.

In this work, we identified the chaperone Anterior-gradient protein homolog 2 (AGR2) as an interactor and essential regulator of IRE1β activity. AGR2 is a protein disulfide isomerase (PDI) uniquely expressed in goblet cells and involved in mucus maturation (Park et al, 2009). Perturbations in AGR2 cause intestinal pathology due to disruptions in mucin maturation both in murine models (Park et al, 2009) and in humans (Al-Shaibi et al, 2021). Here, we describe AGR2's novel role as a regulator of IRE1β activity by keeping the protein in an inactive (monomeric) form, which is conceptually similar as to how the chaperone BiP tunes IRE1α activity. In non-goblet cells, deficient for endogenous AGR2, overexpression of IRE1β leads to spontaneous dimerization and unrestrained activation, causing cell death, which can be rescued by co-expression with exogenous AGR2.

## Results

### IRE1β activity is attenuated upon overexpression in goblet cells

Previous studies on the endonuclease activity of IRE1β, have generally relied on cell types that do not represent the natural environment for IRE1β, as IRE1β displays a much more restricted expression pattern compared to IRE1α (Fig. 1A). HeLa cells, where rapid overactivation and even cell death is reported upon exogenous expression of IRE1β (Iwawaki et al, 2001), express only background levels of IRE1β transcript (Fig. 1A). We therefore selected two cell lines with highly diverging endogenous expression levels of IRE1β at transcript and protein levels, to compare IRE1β endonuclease activity and toxicity upon overexpression. Calu-1 cells are lung epithelial cells that do not express IRE1β at endogenous level, like most other cell lines, whereas the goblet cell-like LS174T line does show endogenous expression of IRE1β transcript and protein (Figs. 1A and EV1A). To assess the endonuclease activity of IRE1β specifically without any confounding effects of IRE1α endonuclease activity, both cell lines were engineered by CRISPR-Cas9 for IRE1α deficiency and transduced with a lentiviral, doxycycline-inducible IRE1β expression module (Calu-1$^{ERN1-/-IRE1βFLAG-DOX}$ and LS174T$^{ERN1-/-IRE1βFLAG-DOX}$) (Fig. 1B). As reported previously for HeLa cells, overexpression of IRE1β in Calu-1 cells induced increasing cell death over time (Fig. 1C, top panels at 72 h post doxycycline induction). This process could be reversed by adding the IRE1-specific endonuclease inhibitor 4μ8C (Cross et al, 2012), but occurred independently of IRE1α, as both *ERN1*+/+ and *ERN1*−/− Calu-1 cells exhibited IRE1β-mediated toxicity that could be rescued by 4μ8C (Fig. EV1B). All further experiments were performed with *ERN1*−/− cell lines. In contrast to our observations in the lung epithelial Calu-1 cells, overexpression of IRE1β in colon epithelial LS174T cells did not cause cellular toxicity (Fig. 1C, lower panels), demonstrating that the cellular context strongly influences IRE1β behavior. The difference in phenotype was not caused by a lack of transgene expression in LS174T cells, as an anti-FLAG-IRE1β western blot demonstrated even higher transgene expression levels in LS174T cells compared to Calu-1 cells (Fig. 1D), which was consistent over time (Fig. EV1C). Notably, in the presence of 4μ8C, Calu-1 cells expressing high levels of IRE1β could be readily retrieved (Fig. EV1C).

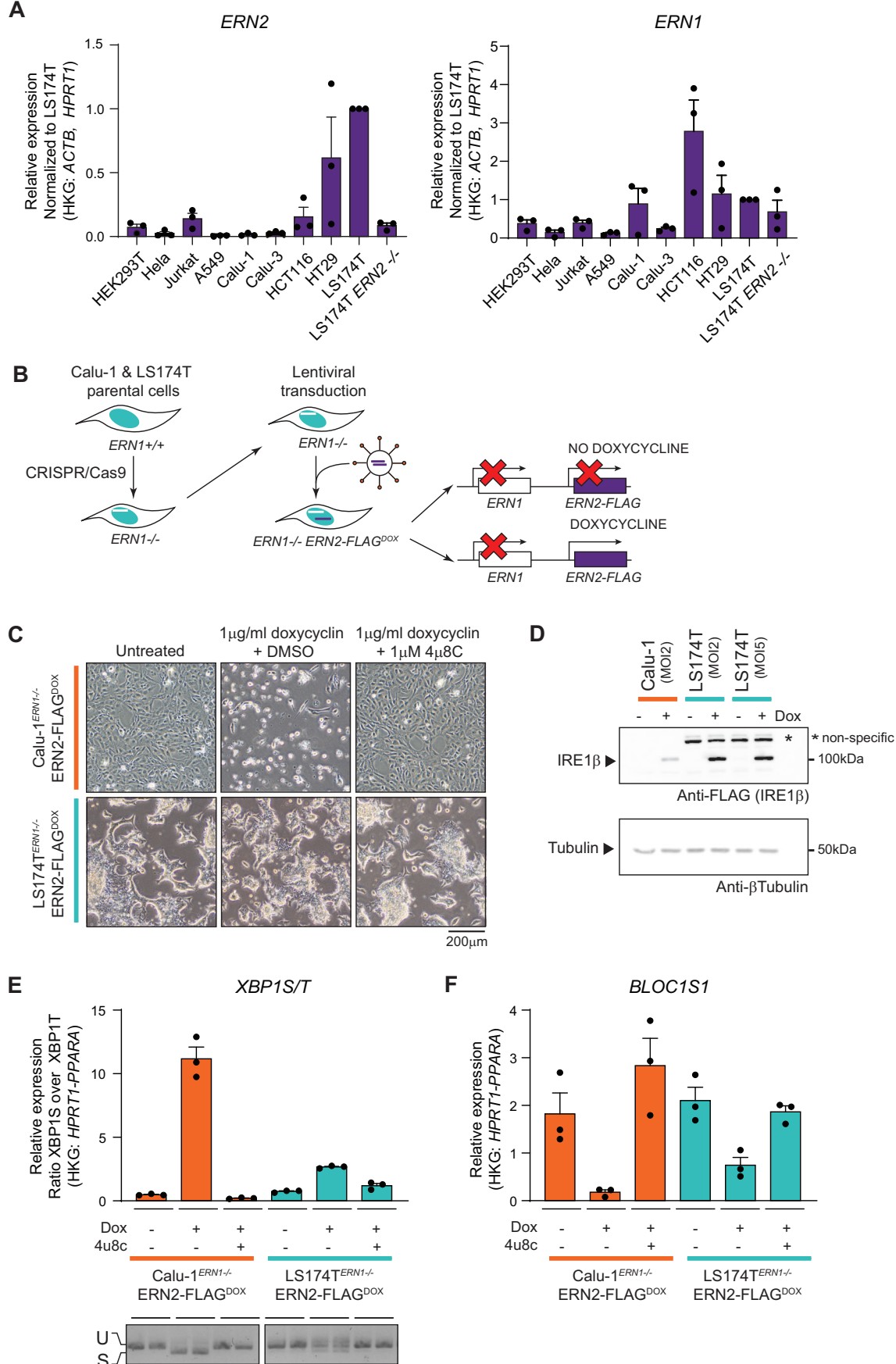

Figure 1. IRE1β activity is attenuated upon overexpression in goblet cells.

(A) RT-qPCR analysis of *ERN2* (IRE1β) and *ERN1* (IRE1α) transcript expression in human cell lines. $N = 3$ culture dishes were sampled and transcript levels are shown relative to the expression in LS174T parental cells. Error bars show SEM. (B) Schematic representation of cell lines used for further studies. First, *ERN1*−/− clones of the LS174T and Calu-1 parental lines were established by CRISPR/Cas9. Then, *ERN1*−/− cells were transduced with a TetOn module for doxycycline-controlled IRE1β-FLAG expression. (C) Photographs showing the phenotype of cultures overexpressing IRE1β-FLAG. Left panels show untreated cultures, middle panels show cultures that received 1 μg/ml doxycycline for 72 h (and thus express IRE1β-FLAG), cultures in the right panels received 1 μg/ml doxycycline and 1 μM IRE1 endonuclease inhibitor 4μ8C. Scale bar represents 200 μm. Representative of three independent experiments. (D) Western blot verifying transgene expression in the cell lines used in (C). All remaining adherent cells after 24 h of transgene induction were collected and lysates were probed for FLAG-IRE1β expression via immunoblot. Tubulin was used as a loading control. Multiplicity of Infection (MOI) indicates the theoretical number of viral particles added. (E, F) RT-qPCR analysis of *XBP1S/T* (E) and *BLOC1S1* transcript levels (F) after 24 h of transgene induction. Representative of three independent experiments with three replicates per condition. Error bars show SEM. (E) bottom picture shows XBP1 splicing in the same samples assayed by conventional PCR, representative of two independent experiments. The fast-migrating band is the spliced XBP1 transcript, and the slow-migrating band the XBP1 unspliced transcript. Source data are available online for this figure.

To measure spontaneous IRE1β endonuclease activity more precisely, XBP1 splicing was quantified in cultures overexpressing IRE1β for 24 h. Also in this case, LS174T cells exhibited a severely diminished activity of IRE1β, with Calu-1 cells reaching much higher levels of XBP1 splicing according to qPCR and PCR analyses (Fig. 1E). Finally, RIDD activity monitored via the degradation of the canonical RIDD target *Bloc1s1* was less induced upon overexpression of IRE1β in LS174T cells compared to Calu-1 cells, albeit not fully disrupted (Fig. 1F). Overall, these data demonstrate that upon overexpression, IRE1β displays basal activity that is repressed in the context of a goblet cell-like cellular background.

## The mucin chaperone AGR2 is a goblet cell-specific interactor of IRE1β

Higher XBP1 splicing levels in Calu-1 cells compared to LS174T cells could be explained by a regulating factor present in the latter (which, like goblet cells normally express IRE1β), but absent in cell types like Calu-1 (that do not normally express IRE1β). To identify potential candidate proteins that could fulfill this role, affinity-purification mass spectrometry (AP-MS) was performed in LS174T cells using both IRE1β-FLAG and IRE1α-FLAG expressed from the doxycycline-inducible system (Fig. 2A). Co-purified proteins were digested and analyzed by mass spectrometry. The enrichment of peptides in IRE1-FLAG samples compared to empty vector control was assessed by label-free quantification (LFQ) and two-sample *t* test. After analysis, 26 proteins had a log$_2$ fold change (log$_2$FC) enrichment over control cells of >2, and log$_{10}$Adj *P* value of >2 in the IRE1α-FLAG pulldown samples compared to control cells (Fig. 2B,C, left panel). For IRE1β-FLAG, 55 proteins remained using the same criteria (Fig. 2B,C, right panel). In total, 11 proteins were commonly identified for both IRE1 paralogues (Fig. 2B, purple intersection) including BiP (*HSPA5*), the SEC61 translocon and CDC37, a kinase chaperone previously identified to interact with IRE1α (Mandal et al, 2007; Ota and Wang, 2012). IRE1β co-precipitated with many ribosomal proteins (in line with IRE1β's previously reported ability to cleave ribosomal RNA (Iwawaki et al, 2001)) and notably, the goblet cell-specific protein disulfide isomerase AGR2 (Fig. 2B, light-blue region). Of all the interaction partners uniquely identified for IRE1β, the chaperone AGR2 stood out as AGR2 had been described as an essential mediator of MUC2 processing and secretion (Park et al, 2009), while IRE1β has been shown to degrade *Muc2* mRNA pools (Tsuru et al, 2013). Furthermore, their interaction yielded one of the highest fold changes at a log$_2$FC of ~6.5 (see Fig. 2C, right panel and Dataset EV1). For further validation of the

specificity of the interaction between IRE1β and AGR2, an antibody-free system was utilized, in which both IRE1 paralogues were biotin-tagged using the Avi-tag (Fairhead and Howarth, 2015), which can only be biotinylated and pulled down by streptavidin beads upon co-expression of the *E. coli* biotin ligase BirA. Also in this overexpression system in HEK293T cells, AGR2 interacted specifically with IRE1β, while only a very faint signal interacting with IRE1α was retrieved (Fig. 2D). Finally, the interaction between endogenous AGR2 and IRE1β was confirmed in colon tissue, using *Agr2*-deficient mice as a negative control (Fig. 2E). In summary, these data corroborate the PDI AGR2 as a specific interactor of IRE1β, not IRE1α.

## Co-expression of AGR2 dampens endonuclease activity of IRE1β

Next, we assessed the endogenous expression levels of AGR2 in the different cell lines we previously analyzed for IRE1β expression (Figs. 1A and EV2A). This revealed that LS174T cells show a prominent expression of AGR2, while Calu-1 cells do not (Fig. EV2A). HeLa cells, which die upon overexpression of IRE1β (Iwawaki et al, 2001), also do not express AGR2 endogenously (Fig. EV2A). Of note, BiP expression levels appeared to correlate with those of IRE1α, while AGR2 expression levels were more closely correlated to the expression levels of IRE1β (Fig. EV2A,B, compare the pattern of blue inserts showing *ERN1* and *ERN2* expression to purple bars showing chaperone expression). This suggests that AGR2 could be a specific modulator of IRE1β signaling in goblet cells by providing protection from unrestrained endonuclease activity and toxicity. Along these lines, other members of the PDI family have been implicated in regulating ER stress responses (Eletto et al, 2016; Eletto et al, 2014; Groenendyk et al, 2014; Higa et al, 2014). To test the hypothesis that AGR2 may regulate IRE1β, we monitored endonuclease activity-dependent cell death in Calu-1 cells upon co-expression of IRE1β and AGR2. Calu-1$^{ERN1-/-IRE1βFLAG-DOX}$ cells were transduced with a lentiviral construct constitutively expressing AGR2 ('Calu-1$^{AGR2}$') or ER-targeted BirA as control protein ("Calu-1$^{Control}$"). Endonuclease activity, as measured by XBP1 splicing, was severely attenuated in Calu-1$^{AGR2}$ cells compared to Calu-1$^{Control}$ cells (Figs. 3A and EV2C,D). In addition, AGR2 expression dampened degradation of the canonical RIDD target *Bloc1s1* (Fig. 3B). Of note, co-expression with AGR2 almost completely reversed the cellular toxicity triggered by IRE1β expression (Fig. 3C) as quantified by Annexin V/Live-Dead staining via flow cytometry in cells either expressing or lacking AGR2 (Figs. 3D and EV2E). After doxycycline-induced expression

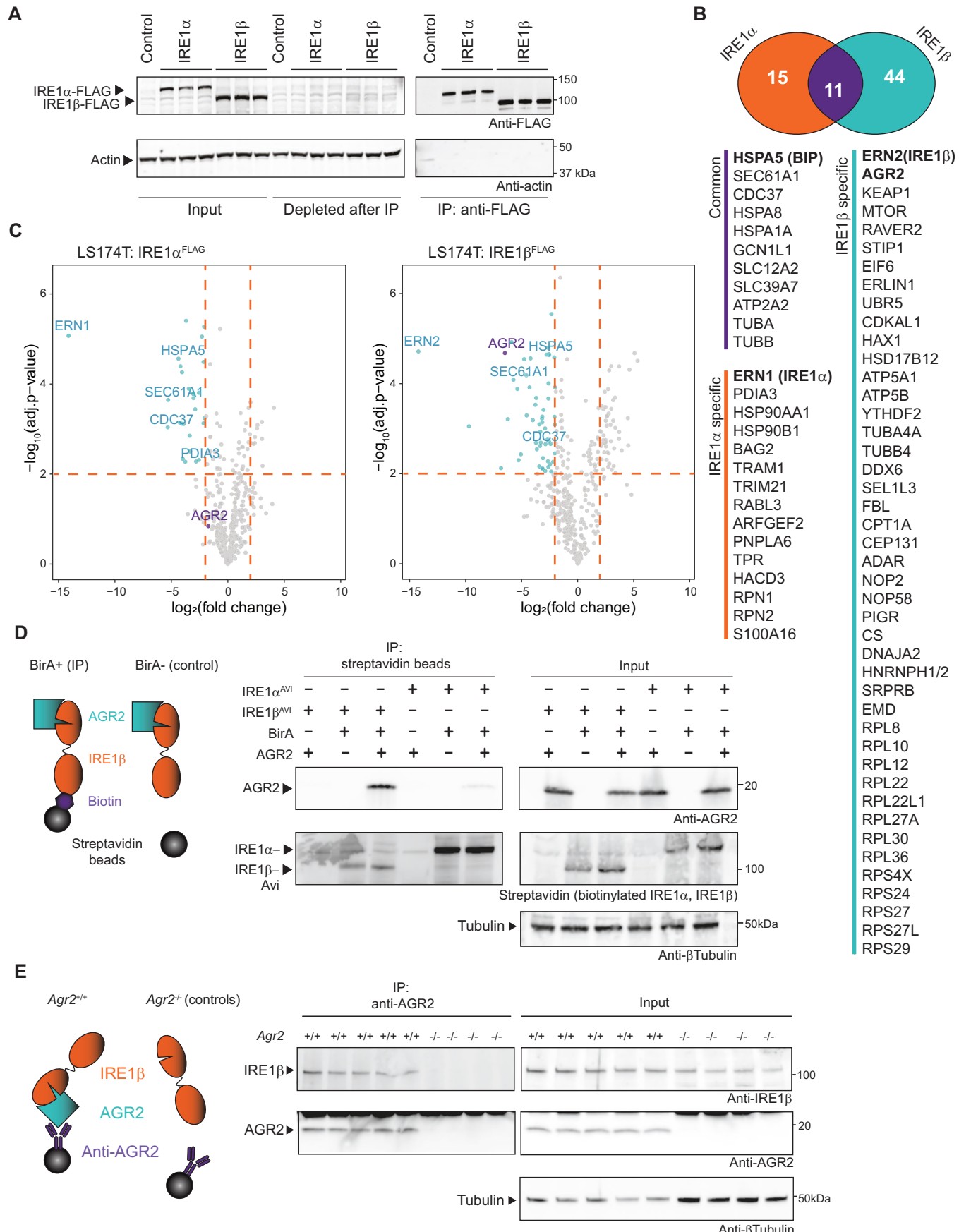

◀  **Figure 2.   The mucin chaperone AGR2 is a goblet cell-specific interactor of IRE1β.**

(A) Verification of transgene expression and successful immunoprecipitation of FLAG-tagged IRE1 in the samples analyzed by MS in (B, C). Lysates were probed for IRE1-FLAG expression using anti-FLAG, and actin was used as a loading control. (B) Proteins with a log$_2$FC enrichment of >2 and log$_{10}$Adj $P$ value of >2 for IRE1α-FLAG and IRE1β-FLAG immunoprecipitation (IP) compared to control cells. The Venn diagram shows the number of proteins that were detected uniquely associated with one of the two IRE1 paralogues or that were commonly identified with both IRE1 paralogues. (C) Volcano plot depicting the cutoff criteria and significantly enriched proteins in each IP. $X$ axis shows the log$_2$ fold change of the measured peptide intensities of a given protein in the control condition over the IRE1 IP condition. $Y$ axis shows the FDR corrected $P$ value obtained by two-sample $t$ test in Perseus. (D) Confirmation of specific interaction between AGR2 and IRE1β, but not IRE1α. IRE1 proteins were tagged with an Avi-tag that is specifically biotinylated upon BirA co-expression. The biotinylated Avi-tag was precipitated using streptavidin beads. For control conditions, BirA was omitted. Blots were probed for co-precipitation of AGR2 and streptavidin to detect Avi-tag-biotinylated IRE1. Tubulin was used as a loading control. Non-specific signal is indicated with an asterisk. Representative of two independent experiments. (E) Confirmation of the AGR2-IRE1β interaction in murine tissue. Colons were isolated and digested, and IP was performed using anti-AGR2. Agr2-deficient mice were used as a negative control to assess whether IRE1β binds specifically to the antibody/bead complex. IP samples were probed for IRE1β co-precipitation via immunoblot. Tubulin was used as a loading control. Representative of two independent experiments.
Source data are available online for this figure.

of IRE1β, protein levels in surviving cells were analyzed by SDS-PAGE (Fig. 3E). This revealed that any remaining Calu-1 cells showed very low IRE1β expression levels (lane 2), unless 4μ8C was added to the cultures to block IRE1β endonuclease activity (lane 3), in line with the notion that none of the IRE1β-expressing cells survived in the culture dish when IRE1β activity was not inhibited. Upon co-expression of AGR2, IRE1β expressing cells could be readily retrieved (lanes 5–6), and inhibition of endonuclease activity by 4μ8C treatment only led to a marginal increase in the amount of IRE1β that was retrieved (Fig. 3E).

On the other hand, when endogenous AGR2 was (partially) depleted from LS174T cells via siRNA (Fig. 3F), XBP1 splicing was elevated, which was apparent both at mRNA (Fig. 3G) and at protein level (Fig. 3H,I) and could be reversed by co-treatment with 4μ8C (Fig. 3G–I). Of note, 4μ8C also inhibited the basal XBP1 splicing levels in LS174T cells in non-targeting siRNA control conditions, showing that even in the presence of endogenous AGR2 levels, IRE1β retained some baseline activity (Fig. 3G). In summary, these data establish AGR2 as a direct modulator of IRE1β endonuclease activity in goblet cells.

## AGR2 blocks IRE1β activity through disruption of IRE1β oligomerization

We next addressed how AGR2 mediates IRE1β inhibition. Previously, PDIA6 and PDIA1, two other members of the large family of PDI's, have been implicated in reversing IRE1α oligomerization (Eletto et al, 2014; Yu et al, 2020), and we hypothesized that AGR2 could operate in a similar manner. To investigate this possibility, gel filtration assays were set up in two different cell lines, HEK293T cells or Calu-1$^{ERN1-/-}$ cells (Fig. 4A), in which the oligomerization status of overexpressed IRE1β could be monitored. The HEK293T cells were engineered to transiently overexpress an IRE1β construct fluorescently tagged with mono-meric superfolder GFP (msfGFP). The Calu-1$^{ERN1-/-}$ cells were stably expressing a doxycycline-inducible IRE1β-FLAG protein (Fig. 4A). IRE1β-msfGFP eluted in 2 peaks: a large peak at ~14.2 mL and a smaller peak at ~15.3 mL (Fig. 4B, dotted lines). These two species shifted in intensity when AGR2 was co-expressed: the first peak shifted to ~14.5 mL and became less abundant, while the peak at ~15.3 mL became more prominent (Fig. 4B, orange and purple traces). Based on the previously established elution profile of IRE1β in dodecylmaltoside (Grey et al, 2020), the later elution fraction most likely represented monomeric IRE1β-msfGFP, while the larger fraction most likely represented

dimeric IRE1β-msfGFP, indicating that co-expression of AGR2 shifted the IRE1β equilibrium towards the monomeric forms. When cleared protein lysates of Calu-1$^{Control}$ and Calu-1$^{AGR2}$ cells were fractionated by gel filtration into 0.2 ml fractions, similar findings were obtained (Fig. 4C). The early eluting fraction remained present even upon AGR2 co-expression, but a late-eluting (most likely monomeric) fraction of IRE1β-FLAG could only be observed in Calu-1$^{AGR2}$ cells, but not in Calu-1$^{Control}$ cells. Upon IRE1β expression, AGR2 preferentially co-eluted with these smaller molecular weight fractions but was absent from these fractions when IRE1β was not expressed, again indicating that the interaction of AGR2 with IRE1β appeared to disrupt IRE1β dimers and/or oligomers (Fig. 4C). While gel filtration is generally feasible for assessing shifts in IRE1 complex sizes, we cannot deduce the precise molecular weight of the IRE1β complex, as many other cellular proteins may bind IRE1β when using whole cell lysates. In addition, during gel filtration, a small sample is diluted in a larger column volume, which could affect the efficiency of protein complex formation between a membrane-based protein such as IRE1β and an ER-luminal protein like AGR2. We therefore explored an alternative setup to assess whether AGR2 could disrupt IRE1β dimers. To this end, a co-immunoprecipitation experiment was designed in which IRE1β was expressed fused to either the Avi-tag or a FLAG-tag (Fig. 4D). Efficient co-precipitation of IRE1β-FLAG species with the biotinylated Avi-tagged IRE1β bait indicated the presence of at least IRE1β dimers. Co-expression of AGR2 led to a disruption of the IRE1β-dimers as observed by a reduction of FLAG-IRE1β signal co-precipitating with Avi-IRE1β in favor of AGR2 co-precipitation (Fig. 4E, compare lanes 2 and 3). This indicated that AGR2 was able to shift IRE1β complexes that were at least dimeric in size to a monomeric state. This molecular transition was dose-dependent, with a high excess of AGR2 over FLAG-IRE1β having the strongest effect (Figs. 4F and EV3). Together with the data provided in the manuscript by Neidhardt et al (2023), these findings revealed that AGR2 expression disfavored the formation of IRE1β dimers.

## The catalytic-dead C81S and disease-causing H117Y mutations in AGR2 abrogate its ability to bind and inhibit IRE1β activity

In literature, several key residues have been identified in AGR2 that impact its function and/or tertiary structure (Fig. 5A). As AGR2 is characterized by an unusual CXXS thioredoxin catalytic motif, a single point mutation (C81S) is sufficient to disrupt its thiol

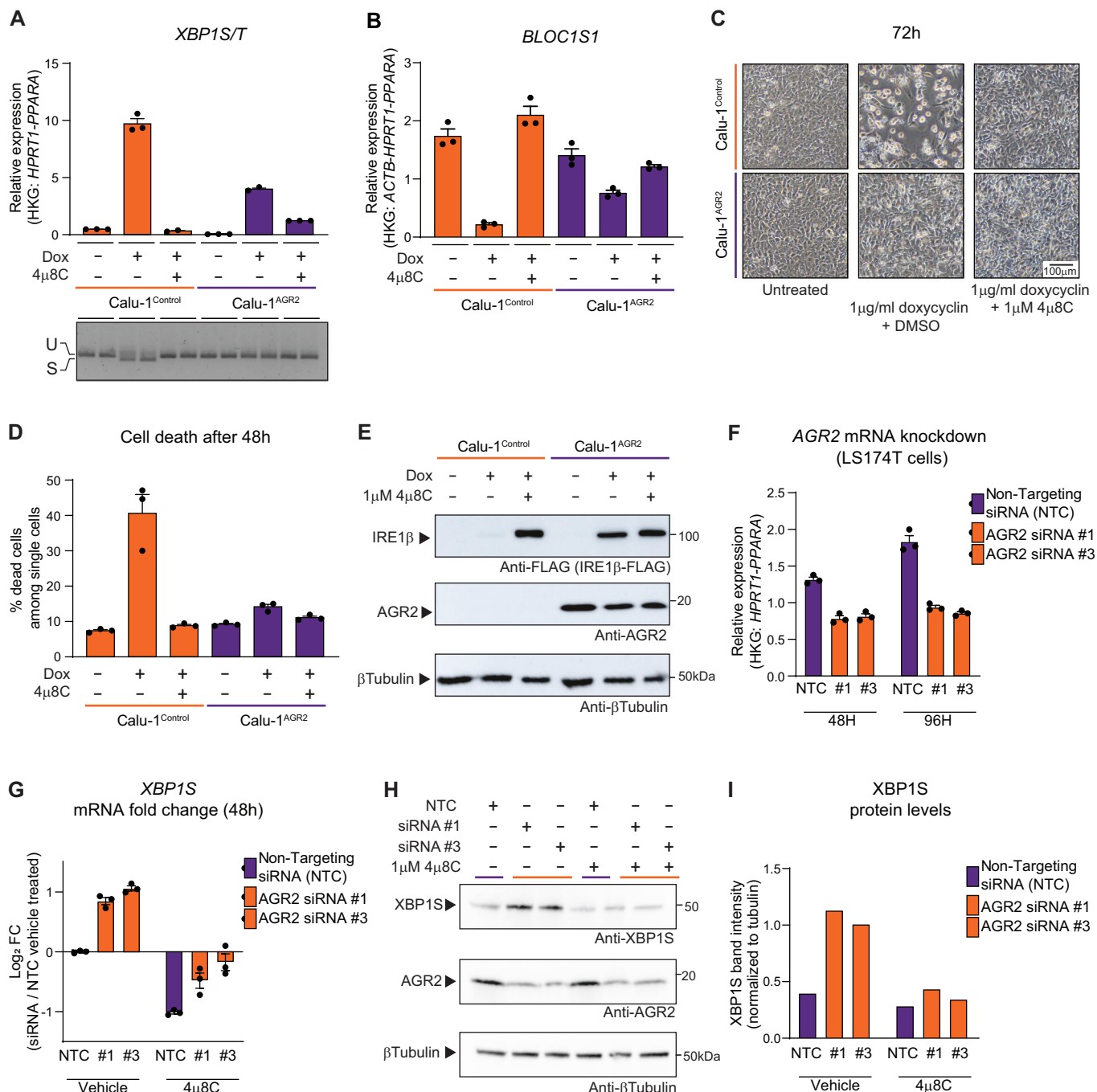

exchange (Park et al, 2009). The nuclear magnetic resonance (NMR) structure of AGR2 revealed that AGR2 exists in a dimer, and that the single amino acid substitution mutations E60A and K64A behave as obligate monomers (Patel et al, 2013). Recently, the amino acid substitution H117Y has been identified as disease-causing in families where homozygous individuals display severe early-onset inflammatory bowel disease (IBD) (Al-Shaibi et al, 2021; Bertoli-Avella et al, 2022). We wondered whether AGR2 requires dimerization and its thioredoxin motif to mediate the interaction with IRE1β. To this end, all the mutations were assayed in a similar co-IP setup as performed in Fig. 4E,F, to assess simultaneously whether an AGR2 mutant retained binding to

IRE1β and/or the ability to disrupt IRE1β dimers (Fig. 5B). This revealed that the monomeric E60A and K64A mutants retained the capacity to bind IRE1β and consequently, to disrupt the formation of IRE1β-Avi/IRE1β-FLAG dimers. On the other hand, both the C81S and H117Y mutations lost interaction with IRE1β, which was reflected by a sustained formation of IRE1β-Avi/IRE1β-FLAG dimers, similar as in conditions without exogenous added AGR2 (Fig. 5C). Quantification of these experiments confirmed the inverse relationship between AGR2 expression levels and the ability of IRE1β to dimerize. Co-expression with AGR2 mutants that retained binding to IRE1β (AGR2 WT, E60A, K64A), led to a decrease in IRE1β dimer/oligomer formation (Fig. 5D). Conversely,

◄

**Figure 3. Co-expression of AGR2 dampens endonuclease activity of IRE1β.**

Calu-1$^{ERN1-/-IRE1βFLAG-DOX}$ cells were transduced with a constitutive AGR2 transgene ("Calu-1$^{AGR2}$"). Cells denoted as "Calu-1" are the original Calu-1$^{ERN1-/-IRE1βFLAG-DOX}$ cells. (A) RT-qPCR analysis of *XBP1S*/T transcript levels after 24 h of transgene induction using 1 µg/ml doxycycline. Bottom picture shows XBP1 splicing in the same samples assayed by conventional PCR. Representative of four independent experiments with three replicate wells per condition. Error bars show SEM. (B) RT-qPCR analysis of *BLOC1S1* transcript levels after 24 h of transgene induction using 1 µg/ml doxycycline. Representative of five independent experiments with three replicates per condition. Error bars show SEM. (C) Photographs showing the phenotype of cultures overexpressing IRE1β-FLAG with and without exogenous expression of AGR2. Left panels show untreated cultures, middle panels show cultures that received 1 µg/ml doxycycline for 72 h, and cultures in the right panels received 1 µg/ml doxycycline and 1 µM 4µ8C. Scale bar represents 100 µm. Representative of three independent experiments. (D) Quantification of cell death in cultures overexpressing IRE1β-FLAG with and without exogenously added AGR2 after 48 h. All cells in the culture dish were stained with Annexin V and Live/Dead stain and analyzed by flow cytometry. All single and double positive cells were considered as dead cells. Representative of two independent experiments with three replicate wells per condition. Error bars show SEM. (E) Analysis of IRE1β-FLAG and AGR2 expression in the cell lines used for (C). Only cells that remained attached in the dish were collected and lysates were probed for IRE1β expression using anti-FLAG and AGR2 expression using anti-AGR2. Tubulin was used as a loading control. (F) RT-qPCR validation of AGR2 knockdown efficiency in LS174T$^{ERN1-/-IRE1βFLAG-DOX}$ cells. NTC is a non-targeting control pool of siRNA's, #1 and #3 are siRNA's targeting AGR2. Representative of three independent experiments with $n = 3$ replicate wells per condition. Error bars show SEM. (G) Log$_2$ fold changes in XBP1 splicing after AGR2 partial knockdown and/or treatment with 4µ8C or DMSO (vehicle). Splicing is shown as a log$_2$ fold change of XBP1S mRNA over the NTC/vehicle-treated cells. Representative of three independent experiments with $n = 3$ replicate wells per condition. Error bars show SEM. (H) Western blot confirmation of (F, G). Proteins were extracted after 72 hours and probed for XBP1S, AGR2 and tubulin expression. Representative of two independent replicates. (I) Quantification of XPP1S protein levels normalized to tubulin in $n = 2$ experiments represented in (H). Source data are available online for this figure.

co-expression with AGR2 mutants that lost binding to IRE1β (C81S, H117Y), led to an increase in IRE1β dimer/oligomer formation (Fig. 5D).

Based on the observation that WT AGR2 attenuates IRE1β-mediated cytotoxicity, we speculated that AGR2 mutations that block binding of AGR2 to IRE1β would unleash IRE1β endonuclease activity. To assess this hypothesis, all AGR2 mutants were transduced as constitutive lentiviral constructs in the Calu-1$^{ERN1-/-IRE1βFLAG-DOX}$ cell line. All lines show similar expression of AGR2 protein (Fig. EV4). The capability of each mutant to disrupt the catalytically active IRE1β dimer (Fig. 5C,D) correlated with their ability to inhibit IRE1β-mediated cell death (Fig. 5E,F) and endonuclease activity as measured by XBP1 splicing (Fig. 5G) and BLOC1S1 RIDD activity (Fig. 5H). This demonstrates that the inhibitory effect of AGR2 on IRE1β endonuclease activity and resulting cytotoxicity was strictly dependent on its ability to efficiently interact with IRE1β. Interestingly, even though the C81S mutation failed to block cellular toxicity, some inhibitory effect on IRE1β endonuclease activity could still be observed (Fig. 5G,H). This was in contrast to the H117Y mutation, which completely lost its ability to modulate endonuclease activity, indicating that the H117 residue plays a more critical role compared to C81 in regulating IRE1β activity.

Overall, our data establish AGR2 as a goblet cell-specific regulator of IRE1β endonuclease activity by shifting the protein towards a monomeric, catalytically inactive state. In cells that do not express AGR2, overexpression of IRE1β leads to unrestrained endonuclease activity and cell death.

## Discussion

Goblet cells, embedded in the epithelia lining the mucosal interfaces of the respiratory and GI track, are specialized secretory cells involved in the production and secretion of the gel-forming mucins MUC5A/B and MUC2, the major constituents of mucus. The proper folding and assembly of the 5100 aa MUC2 proteins is a real "tour-de-force", following a highly structured process involving the formation of multiple disulfide bridges at the C- and N-termini, dimerization and trimerization steps in the ER and Golgi,

respectively, and massive O-glycosylation of the central Proline, Threonine and Serine (PTS) domain (Fass and Thornton, 2023). This specialized function is reflected by the goblet cell transcriptomic and proteomic profile, which highlights activation of cellular pathways associated with protein production, folding, vesicle transport, glycosylation and secretion (Nyström et al, 2021). Some of these genes, such as *Clca1, Zg16, Fcgbp1, Ern2, Agr2* or the transcription factors *Atoh1* or *Spedf* appear uniquely expressed in goblet cells (Haber et al, 2017). Amongst them, the protein disulfide isomerase (PDI) AGR2 is one of the most abundantly expressed proteins in the goblet cell ER (Nyström et al, 2021).

AGR2 is a highly conserved protein, originally identified in Xenopus, where it was shown to play an essential role in development (Aberger et al, 1998; Bradley et al, 1996; Sive et al, 1989). It has an atypical KTEL ER-retention motif that localizes the protein to the ER (Gupta et al, 2012), however, it has been found to be secreted as well (Bergstrom et al, 2014; Johansson et al, 2009). Several observations suggested a role for AGR2 in mucus homeostasis. Polymorphisms in the *AGR2* gene have been associated with increased risk of Ulcerative colitis and Crohn's disease (Zheng et al, 2006) and *Agr2-/-* mice lack an inner mucus layer in their GI tract (Bergstrom et al, 2014; Park et al, 2009). An earlier study suggested that AGR2 would use its isomerase activity to form mixed disulfide bonds with MUC2, in this way assisting in mucus folding (Park et al, 2009). However, this could not be confirmed by others (Bergstrom et al, 2014) and up to now it remains unclear how AGR2 contributes to mucus homeostasis. As mentioned, the protein has been retrieved as an integral component of the intestinal mucus layer, indicative of a potential extracellular function as well (Johansson et al, 2009).

We here identified AGR2 as a specific regulator of IRE1β, a sensor of the UPR that is uniquely expressed in goblet cells, and that plays an essential role in goblet cell development and mucus homeostasis (Cloots et al, 2021b; Grey et al, 2022; Tsuru et al, 2013). AGR2 interacts specifically with IRE1β, but not with its ubiquitously expressed paralogue IRE1α, both upon overexpression in cell lines and in vivo in mouse colon tissue. As shown by Neidhardt et al in a parallel manuscript, this interaction happens in a direct manner. Binding of AGR2 to IRE1β dampens its endonuclease activity, leading to a reduction in XBP1s splicing

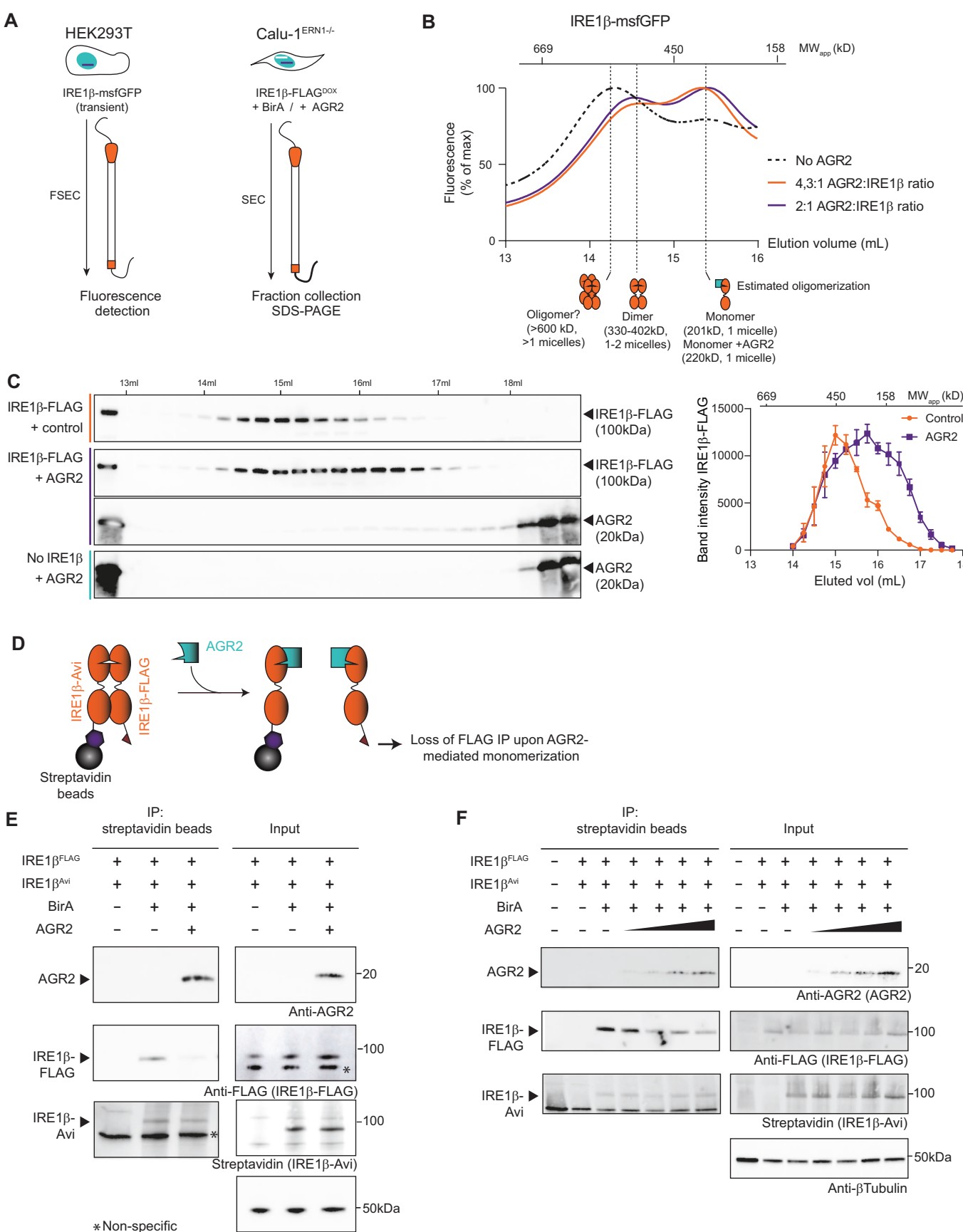

**Figure 4.   AGR2 blocks IRE1β activity through disruption of IRE1β oligomers.**

(A) Schematic overview of gel filtration experiments in (B, C). (B) msfGFP fluorescence measured during elution of HEK293T lysates overexpressing IRE1β in the absence of AGR2 (black dotted trace) or in the presence of AGR2 (orange and purple traces indicating different ratios of transfected AGR2:IRE1β plasmid). Top scale represents the approximate elution profile and expected MW of protein standards. Bottom drawings indicate expected oligomerization status based on protein standards and the previously obtained elution profile (Grey et al, 2020). Each trace shows a single chromatography run. (C) IRE1β-FLAG expression in fractions collected after gel filtration of protein lysates from Calu-1ERN1-/-IRE1βFLAG-DOX cells, in the absence or presence of additional AGR2 expression. Line graph shows quantification of band intensities from the gel of three independent replicates. Error bar shows SEM. (D) Schematic representation of competition IP experiments in (E, F). IRE1β is expressed with an Avi-tag or FLAG-tag in equimolar amounts. After biotinylation of the Avi-tag by BirA, both the Avi-tag and FLAG-tag will be detected after streptavidin IP if dimers have been formed. If addition of another protein (e.g., AGR2) would block this process, a loss of signal is expected. (E) Competition IP showing loss of dimer formation upon co-expression of AGR2. Samples were immunoblotted with anti-AGR2, anti-FLAG and Streptavidin. Tubulin was used as a loading control in input samples. Representative of three independent experiments. Non-specific signal is indicated with an asterisk. (F) Competition IP demonstrating concentration-dependent loss of dimer formation upon increasing AGR2 co-expression. Samples were immunoblotted with anti-AGR2, anti-FLAG and Streptavidin. Tubulin was used as a loading control in input samples. Representative of two independent experiments. Source data are available online for this figure.

activity and RIDD (Neidhardt et al, 2023). Besides, AGR2 also protects cells from IRE1β-mediated cytotoxicity, a hitherto unexplained phenomenon observed when IRE1β is exogenously expressed in non-goblet cell lines (Iwawaki et al, 2001). Co-expression with AGR2 rescues cells from IRE1β-mediated cell death, in a manner that strictly depends upon their interaction, as any AGR2 mutant that loses interaction with IRE1β also loses its capacity to regulate its endonuclease output. On the reverse, siRNA-mediated reduction in AGR2 expression in goblet cells leads to increased IRE1β activity and XBP1 splicing. At the mechanistic level, we could demonstrate that AGR2 tends to disrupt IRE1β dimers, shifting IRE1β from a dimeric/oligomeric, highly active state towards a monomeric, inactive state. This was also confirmed by using recombinant proteins (Neidhardt et al, 2023) and is conceptually highly reminiscent of how the Hsp70 chaperone BiP regulates IRE1α activity (Bertolotti et al, 2000).

Currently, two alternative mechanisms have been put forward as to how BiP would engage the IRE1α luminal domain (LD), either in an ATP-dependent manner via its substrate-binding domain as a true chaperone–substrate interaction (Amin-Wetzel et al, 2017), either in an ATP/chaperone-independent manner via its nucleotide-binding domain (Kopp et al, 2019). In the first model, unfolded proteins would compete with IRE1 LD for binding to the BiP substrate-binding domain (SBD) (Amin-Wetzel et al, 2017, Preissler and Ron, 2019). In the second model, binding of unfolded proteins to the SBD of BIP would allosterically trigger the release of BiP from IRE1 (Adams et al, 2019; Kopp et al, 2019; Preissler and Ron, 2019). Independent of the upstream mechanism, both models would predict that the dynamic binding of BiP to IRE1α would establish a pool of inactive monomeric IRE1, against its intrinsic propensity to form dimers (Preissler and Ron, 2019). Based on emerging data, BiP does not function isolated, but in cooperation with other chaperones such as members of the protein J family (Amin-Wetzel et al, 2017). In addition to BiP, also members of the large protein disulfide isomerase (PDI) family have been shown to regulate the activity of UPR sensors. The ER-luminal oxidoreductase PDIA6 binds to oxidized Cysteine 148 in the luminal domain of IRE1α, which forms a disulfide bridge with Cys332 during activation-induced oligomerization (Liu et al, 2003). Reduction of the disulfide bridge by PDIA6 promotes inactivation of IRE1α and loss of PDIA6 leads to prolonged IRE1 activity with deleterious effects in *C. elegans* (Eletto et al, 2014). Of note, a later study in mammals observed the opposite and proposed that PDIA6 would sustain long-term activation of IRE1 by an as-yet undefined mechanism (Groenendyk et al, 2014). Finally, also

(phosphorylated) PDIA1 has been shown to attenuate IRE1α activity by binding its luminal domain, this time in a cysteine-independent manner (Yu et al, 2020).

Neither Cys148, nor Cys332 are conserved in IRE1β (Fig. EV5A), but IRE1α Cys109 is conserved in the IRE1β luminal domain at position Cys117 (Zhou et al, 2006). A second cysteine is present in the IRE1β luminal domain (Cys204), but, based on predictive modeling, neither cysteine appears to be solvent accessible in IRE1β. Furthermore, also AGR2 functions differently from canonical PDIs in the sense that it harbors only one single thioredoxin CXXS motif. It has been postulated to form dimers, that could allow catalytic activity (Patel et al, 2013). We found that the interaction of AGR2 with IRE1β occurred independently of AGR2 dimerization but depended on the presence of an active catalytic cysteine, as reflected by a loss of AGR2^C81S capacity to repress IRE1β cytotoxic activity. Similar findings are presented in an accompanying manuscript by Neidhardt et al (2023). While the interaction between IRE1β and AGR2^C81 mutants was weakened (though not fully disrupted), gel filtration assays on recombinant proteins demonstrated a diminished potency of AGR2 cysteine mutants to disrupt IRE1β dimers, in line with our own findings. Still, we can reasonably assume that AGR2 does not engage IRE1β through a mixed disulfide bond. Consistent with this notion, AGR2 was identified as an interactor of IRE1β in an immuno-affinity screen performed under reducing conditions, in which mixed disulfides would not be detected. Neither was the formation of a mixed disulfide between IRE1β and AGR2 observed upon co-immunoprecipitation of the two recombinant proteins (Neidhardt et al, 2023). This suggests that the catalytic cysteine would rather mediate an allosteric effect. To get more insight into how these mutations might affect the interaction between IRE1β and AGR2, we modeled their interaction using AlphaFold2-Multimer (Fig. EV5B,C). While these models have only predictive value, it was surprising to note that all 5 seed models created by AlphaFold2 docked AGR2 to the same flexible loop of IRE1β (amino acids Ala318 to Leu353) with the AGR2 C81 and the clinically relevant H117 motif (Al-Shaibi et al, 2021) facing the interaction site with IRE1β. This could explain why both mutants lost interaction with IRE1β. On the other hand, the C81 moiety in AGR2 appeared to be spatially separated from the two cysteines present in the luminal domain of IRE1β, again suggesting an allosteric rather than a true thiol exchange role for C81. Finally, the two known amino acids involved in AGR2 dimerization (E60, K64) were located at the opposite site of the interaction motif with IRE1β, consistent with our immunoprecipitation experiments. Notably, BiP has been

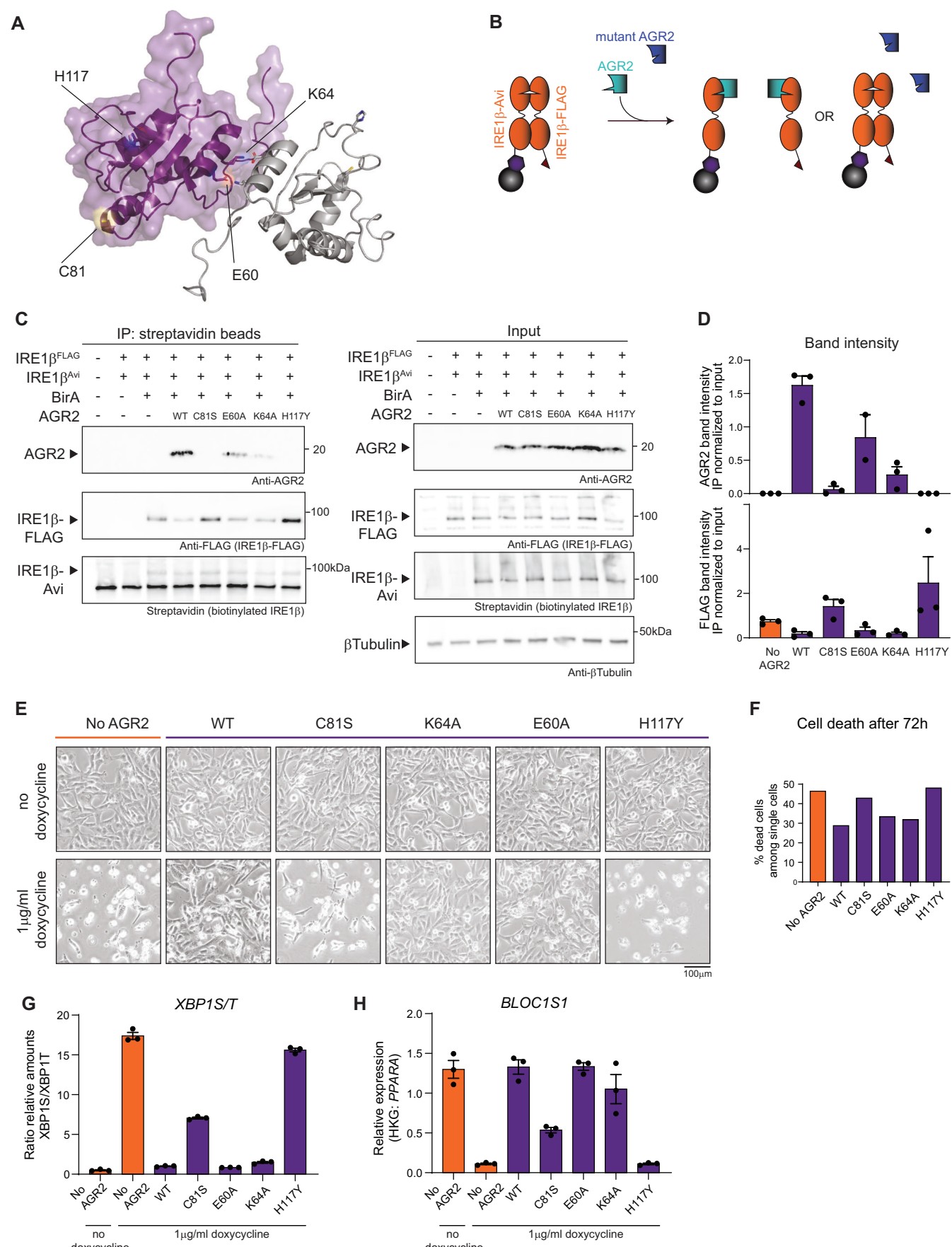

**Figure 5. The catalytic-dead C81S and disease-causing H117Y mutations in AGR2 abrogate its ability to bind and inhibit IRE1β activity.**

(A) The structure of AGR2 (pdb: 2LNS) visualized in PyMol with the relevant mutations indicated. Purple and grey cartoons depict two AGR2 molecules and their dimer structure (Patel et al, 2013), specific residues are represented as ball-and-sticks. (B) Schematic overview of competition IP using AGR2 mutants. (C) Competition IP showing loss of dimer inhibition using C81S and H117Y AGR2 mutants. Samples were immunoblotted with anti-AGR2, anti-FLAG-IRE1β and Streptavidin. Tubulin was used as a loading control in input samples. Representative of three independent experiments. (D) Quantification of IP band intensities normalized over corresponding input sample on western blots from (C), with each data point representing one experiment. Error bars show SEM. (E) Photographs showing Calu-1$^{ERN1-/-IRE1βFLAG-DOX}$ cultures, transduced with a constitutive AGR2 transgene (wild-type or the indicated mutants). IRE1β-FLAG overexpression was induced with 1 μg/ml doxycycline for 72 h. Scale bar represents 200 μm. Representative of two independent experiments. (F) Quantification of cell death in cell lines from (E) after 72 h of transgene expression. All cells in the culture dish were stained with Annexin V and Live/Dead stain and analyzed by flow cytometry. All single and double positive cells were considered as dead cells. Representative of two independent experiments with $n = 2$ culture dishes. (G) RT-qPCR analysis of *XBP1S/T* transcript levels after 24 h of transgene induction using 1 μg/ml doxycycline. Representative of two independent experiments with three replicates per condition. Error bars show SEM. (H) RT-qPCR analysis of *BLOC1S1* transcript levels after 24 h of transgene induction using 1 μg/ml doxycycline. Representative of two independent experiments with three replicates per condition. Error bars show SEM. Source data are available online for this figure.

shown to bind to IRE1α at the same loop position, being held by both the flexible loop and the tail region in IRE1α (Amin-Wetzel et al, 2019). Whether and how BiP can contribute to regulating IRE1β activity remains an open question. Our own AP-MS data are in line with earlier studies showing interaction of BiP with IRE1β, though interaction between BiP and IRE1β appears weaker than its interaction with IRE1α (Bertolotti et al, 2000; Oikawa et al, 2012). In addition, the data in the parallel manuscript by Neidhardt et al conclusively show that recombinant AGR2 is sufficient to block IRE1β dimerization independent of BiP binding and that BiP depletion alone does not activate IRE1β (Neidhardt et al, 2023). All these data combined suggest that BIP and AGR2 might show optimized affinities for the different flexible loop regions of IRE1α and IRE1β, respectively, leading to a more selective inhibitory effect. Such a specificity is further supported by the similarities in expression pattern we found for the IRE1α-BIP, and the IRE1β-AGR2 pair.

Why would the goblet cell-specific UPR sensor IRE1β be regulated by a unique chaperone that is specific to goblet cells? The beauty of the UPR system is how it adapts folding load to match folding capacity by ensuring several negative feedback loops in different cellular contexts. One of the most well-known examples is the mammalian UPR sensor PERK, which upon activation due to dissociation of BiP, temporarily halts cap-dependent protein synthesis and in this way prevents further import of unfolded proteins in the ER (Harding et al, 2000; Scheuner et al, 2001; Sood et al, 2000). More recently, an alternative system has been identified that leads to trimming of the available mRNA pool by the endonuclease activity of IRE1, a process termed RIDD (Hollien et al, 2009; Hollien and Weissman, 2006). RIDD is conserved from yeast to men and has been found downstream of both IRE1α and IRE1β (Maurel et al, 2014). Of note, the upstream triggers of RIDD remain poorly understood, as it has often been described in (artificial) conditions of XBP1 deficiency, and/or in conditions of prolonged stimulation with ER stress-inducing agents such as tunicamycin. Hence, its physiological role remains debated. Still, in many cell types RIDD has been shown to target cell type-specific substrates, such as insulin in pancreatic β-cells (Lipson et al, 2008), immunoglobulin in plasma cells (Benhamron et al, 2014; Tang et al, 2018) or components of the antigen presentation machinery in dendritic cells (Osorio et al, 2014), making a cell type-specific and physiological role for RIDD likely. By making use of IRE1β deficient mice, Tsuru et al demonstrated that IRE1β specifically degrades *Muc2* mRNA in goblet cells by RIDD (Tsuru et al, 2013). How this was regulated remained at the time unclear. We now add

AGR2 as a missing piece in this puzzle (Fig. 6). As a goblet cell-specific chaperone, AGR2 might be perfectly suited to detect mucus folding load in the ER and to relay this information to IRE1β through a specific interaction with its luminal domain. This is highly relevant, as so far, the upstream triggers of IRE1β remain largely unknown, though they appear distinct from classical UPR triggers such as tunicamycin or thapsigargin (Cloots et al, 2021b). We speculate that in conditions of high mucus folding load, AGR2 might dissociate from IRE1β, unleashing its propensity to dimerize, as also shown by (Neidhardt et al, 2023, accompanying manuscript). This could trigger IRE1β-mediated RIDD activity, in this way adapting *Muc2* mRNA availability to the MUC2 folding capacity in the ER. Several questions remain though, especially regarding the specific function of AGR2 in mucus folding, and the regulation of the AGR2/IRE1β interaction.

Finally, *AGR2* has recently come into view as a disease-causing gene in a subset of patients displaying IBD-like symptoms, among others. A homozygous H117Y mutation was identified in twins with severe early-onset IBD (Al-Shaibi et al, 2021), and several other families have been identified that display profound inflammatory phenotypes at their mucosal surfaces due to mutations in AGR2 (Bertoli-Avella et al, 2022). As we have demonstrated that AGR2$^{H117Y}$ loses the ability to interact with and block IRE1β, it is reasonable to assume that these patients do not only exhibit disturbed AGR2 chaperone function but may also be characterized by IRE1β hyperactivity (Fig. 6). Currently, it is unknown whether IRE1β hyperactivity causes symptoms in vivo and how this translates to patient care. Our findings highlight the importance of further understanding the AGR2/IRE1β axis in goblet cell physiology and disease.

## Methods

### Cell culture and compounds

Calu-1, HCT116, and HT29 cells were cultured in modified McCoys 5A (Gibco) supplemented with 10% FBS. LS174T and Calu-3 cells were cultured in Minimum Essential Media (MEM) (Sigma) and supplemented with 1× Glutamax (Gibco), 1× NEAA (Gibco), 1× Na-pyruvate (Gibco) and 10% FBS. HEK293T, A549, and HeLa were cultured in Dulbecco's Modified Eagle Medium (DMEM) (Gibco) and supplemented with 10% FBS. Jurkat cells were cultured in Roswell Park Memorial Institute (RPMI) (Gibco), supplemented with 10% FBS. All cells were cultured in a humidified 37 °C incubator at 5% $CO_2$. All cell lines were obtained through ATCC or ECACC and regularly tested for mycoplasma

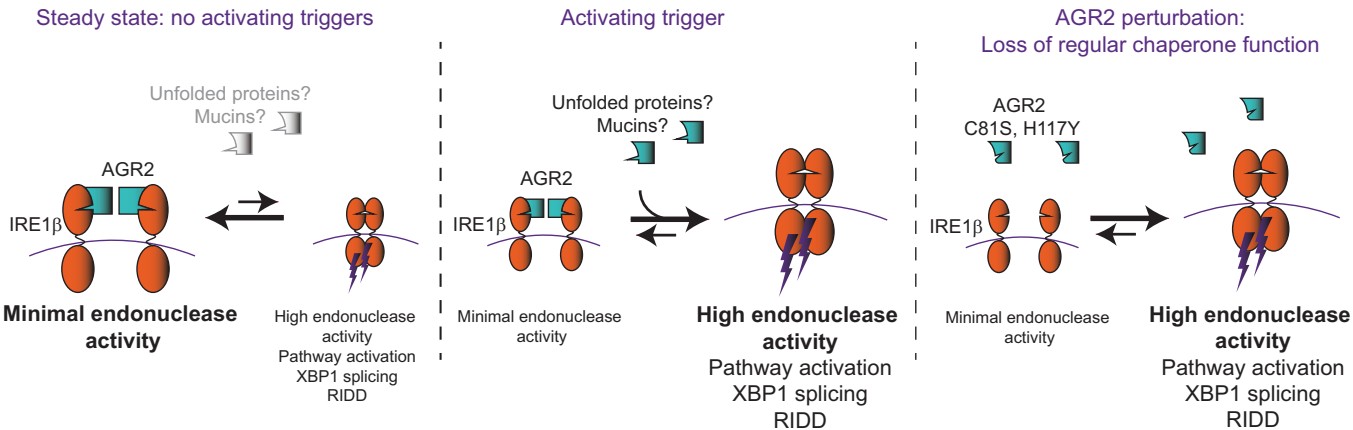

**Figure 6. Proposed mechanism of IRE1β regulation by AGR2.**

In steady-state conditions, AGR2 is bound to most IRE1β molecules and under these conditions, IRE1β is mainly present in the inactive, monomeric forms. As a result, overall IRE1β activity will be low. In conditions where an activating trigger is present (possibly unfolded MUC2 polypeptides, though this remains to be demonstrated), AGR2 is released from IRE1β in favor of binding other AGR2 chaperone substrates. As a result, IRE1β is released, activated and overall IRE1β activity will be high. In case of AGR2$^{C81S}$ and AGR2$^{H117Y}$, interaction with IRE1β is disrupted leading to spontaneous IRE1β dimerization triggering its activity.

contamination. LS174T, A549, HEK293T, and HeLa cells were STR profiled by the VIB-IRC Cell Core Facility.

To induce transgene expression in transduced cell lines (see below), cells were treated with 500 ng–1 µg/ml doxycycline (Sigma-Aldrich) in the culture media for the indicated times. 4µ8C (Sigma-Aldrich) was added at 1–5 µM to inhibit endonuclease activity where indicated.

## CRISPR/Cas9

Calu-1$^{ERN1-/-}$ cells were constructed by transfection with pSpCas9n(BB)-2A-Puro (Feng Zhang lab, Addgene #48139) containing sequences for guides targeting the start codon (WGE ID's 1150311103 and 1150311093). After puromycin selection, cells were seeded as single-cell subclones and clones with no detectable IRE1α protein were treated with tunicamycin to further verify lack of XBP1 splicing in response to tunicamycin. LS174T $^{ERN1-/-}$ cells and LS174T $^{ERN2-/-}$ cells were constructed by transfection with pSpCas9(BB)-2A-GFP (Feng Zhang lab, Addgene #48138) containing sequences for guides targeting exon 8 (WGE ID's 1150303698 and 1150303744 for ERN1, and ID's 1135590972 and 1135590916 for ERN2). After puromycin selection, cells were seeded as single-cell subclones and clones with no detectable IRE1α protein were treated with tunicamycin to further verify lack of XBP1 splicing in response to tunicamycin. For assessing IRE1β expression, clones were genotyped with primers (fwd: 5'-GAGTTGCTGAACAG TGGGGG-3', 5'-GTGTAGACGCCCATCACAGG-3', rev: 5'-GAC CTTACAGGTGCCCCAAG-3') to verify disruption of exon 8 and these clones were further verified by western blot.

## Plasmids

IRE1β-FLAG was ordered through gene synthesis (Gen9), and IRE1α-FLAG was a kind gift from Sarah Gerlo (Ghent University). IRE1-FLAG sequences were amplified with AttB sites, cloned into pDONR223 using BP clonase II (Invitrogen) and transferred

into pDEST51 (Invitrogen) using LR clonase II (Invitrogen). For BirA-Avi co-IP experiments, full-length IRE1α and IRE1β were PCR-amplified with a C-terminal KpnI-SpeI site, flanked by AttB sequences and transferred to pDONR221 with BP clonase II (Invitrogen). The Avi-tag fragment was obtained through annealing of two oligo's (Integrated DNA Technologies) with the correct overhangs (top strand: 5'-CGGCGGTGGCCTGAACGACA TCTTCG AGGCTCAGAAAATCGAATGGCACGAAA-3', bottom strand: 5'-CTAGTTTCGTGCCATTCGATTTTCTGAGC CTCG AAGATGTCGTTCAGGCCACCGCCGGTAC-3') and cloned into the KpnI-SpeI site. The sequences were transferred to pDEST51 or the inducible Virapower TRex expression system (Invitrogen) with LR clonase II (Invitrogen). IRE1β-msfGFP was produced by PCR amplification of IRE1β with a C-terminal KpnI-SpeI site. The msfGFP sequence (Abudayyeh et al, 2017) was ordered through gene synthesis (IDT) and PCR-amplified with KpnI-SpeI cloning sites and cloned into the IRE1β KpnI-SpeI sites. AGR2 was PCR-amplified from LS174T cDNA with AttB sites and transferred in pDONR221 and pLenti-DEST-EF1A-Hygro. All indicated mutations were introduced with the QuickChange protocol (Agilent), and sequences were verified by Sanger sequencing. BirA was PCR-amplified from pDisplay-BirA-ER (Alice Ting lab, Addgene #20856) with AttB sites as described above and transferred to pDEST12.2 or pLenti-DEST-EF1A-Hygro. Plasmid preparations were performed using Nucleobond Midi Xtra (Machery-Nagel).

## Viral vectors and transduction

Lentiviral particles were prepared in HEK293T cells using standard calcium-phosphate transfection of the lentivector with packaging plasmids pMD2.G (Trono lab, Addgene #12259) and pCMV-dR8.74 (Trono lab, Addgene #22036). Viral particles were concentrated through ultracentrifugation. Cell lines were transformed with both the pLenti3.3 repressor at MOI5, the IRE1β-FLAG transgene in

pLenti6.3/TO/DEST at MOI2 and constitutive AGR2 at MOI2 or greater. Calu-1 cells carrying stable integration of the transgene were selected using G418 (0.5 mg/ml), blasticidin (10 µg/ml) and hygromycin (200 µg/ml), respectively. LS174T cells were selected using G418 (0.4 mg/ml), blasticidin (8 µg/ml).

## AP-MS

LS174T cells were plates in 2× 145-cm$^2$ culture dishes/repeat pulldown (a total of three repeats was carried out per conditions). Transgene expression was induced using 1 µg/ml doxycycline. After 24 h, cells were collected in PBS. Cell pellets were washed in PBS, weighed and lysed in 400 µl/100 µg cell pellet AP-MS buffer (50 mM HEPES-KOH pH 8.0, 100 mM KCl, 2 mM EDTA, 0.1% NP-40, 10% glycerol, 1 mM DTT, 0.5 mM PMSF, 0.25 mM sodium orthovanadate, 50 mM β-glycerophosphate, 10 mM NaF and protease inhibitor cocktail (Roche)). Cells were subjected to one freeze-thaw cycle after which lysates were cleared by centrifugation (16,000× $g$, 4DC, 20') and transferred to a new tube where they received BioM2 anti-FLAG (Sigma-Aldrich) loaded Dynabeads MyOne Streptavidin T1 (Invitrogen) (5 µl anti-FLAG with 50-µl beads per combined set of 2× 145-cm$^2$ dish) and were incubated with end-over-end rotation at 4DC for 2 h. Beads were washed, peptides were eluted by on-bead trypsin (Promega) digest and acidified for MS analysis. 2.5 µl sample was injected on a Q Exactive hybrid Quadrupole-Orbitrap analyzer (ThermoScientific). Protein identification was carried out using the Andromeda search engine of the MaxQuant v1.6.3.4. software package (Cox et al, 2014) and the SwissProt human proteome (versions 08-2018). Identified proteins were quantified using label-free quantification and significant enrichment of proteins was determined after imputation of missing values using a Student's paired $t$ test in Perseus v1.5.5.3. Proteins were considered to be enriched when they were characterized by a log$_2$ fold change (log$_2$FC) enrichment of >2, and log$_{10}$Adj $P$ value of >2.

## Co-immunoprecipitation (co-IP)

HEK293T cells were seeded in six-well plates at 5.10$^5$ cells/well and transfected the next day by calcium-phosphate transfection (6.6 µg DNA/transfection) in culture medium containing 1 µM biotin. Cells were transfected with a ~2.5:1 molar ratio of AGR2:IRE1 cDNA (i.e. the AGR2 cDNA fragment was present in a 2.5-fold copy number excess compared to the IRE1β cDNA fragment), except for Fig. 4F, where the cDNA molar ratio ranged as follows: 1:1, 2:1, 3:1, and ~4:1 AGR2:IRE1β cDNA. To mediate biotinylation of the IRE1α or IRE1β-Avi-tag, BirA was included the transfection mix, and replaced by pSV-sport empty vector for control conditions. For competition co-IP, IRE1β-Avi and IRE1β-FLAG plasmids were included in equal molar amounts. Cells were lysed in E1A lysis buffer (1% NP-40, 10% glycerol, 250 mM NaCl, 20 mM HEPES pH 7.9, 1 mM EDTA with cOmplete mini protease inhibitor cocktail and PhosSTOP (Roche)). Cleared lysates were incubated for 1 h at 4DC with Dynabeads Streptavidin T1 (Invitrogen) (8 µl beads per sample). Beads were washed and proteins were eluted directly in loading buffer by incubating beads in loading buffer at 65DC.

## In vivo co-IP

AGR2$^{tm1.2}$Erle/J mice were purchased from the Jackson Laboratory. All mice were housed in SPF conditions under the current EU animal housing guidelines and ex-vivo experiments were approved by the Ethical Committee of the Ghent University Faculty of Science. Mice were euthanized by $CO_2$ asphyxiation, and colons were extracted. After two washing steps in PBS to remove fecal matter, colon pieces were transferred to 1xHBSS containing 0.15% DTT to remove mucus. Pieces were then transferred to digestion buffer (RPMI supplemented with DNAse I and Liberase TM), homogenized with GentleMACS C tubes and incubated in a shaking warm water bath for 20 min. At 10 and 20 min, samples were again homogenized with GentleMACS C tubes. Single-cell pellets were lysed in E1A buffer (1% NP-40, 10% glycerol, 250 mM NaCl, 20 mM HEPES pH 7.9, 1 mM EDTA with cOmplete mini protease inhibitor cocktail (Roche) and PhosSTOP (Roche)) and cleared by centrifugation. Anti-AGR2 antibody (clone 6C5, Santa Cruz Biotechnology) was first bound to Dynabeads Protein G (Invitrogen) according to the manufacturer's instructions. Antibody-bead complexes were added to cleared lysates and incubated with rotation for 2 h. Beads were washed and proteins were eluted directly in loading buffer by incubating beads in loading buffer at 65DC.

## Western blot

Protein lysates and IP samples were all separated on 4–20% MiniProtean TGX or Criterion TGX (Bio-Rad) precast SDS-PAGE gels. Proteins were transferred to Amersham HyBond PVDF or nitrocellulose membranes (Cytiva) and revealed using following antibodies: mouse FLAG-M2-HRP (Sigma, A8592, 1/1000), rabbit anti-IRE1β (Invitrogen, PA5-13921, 1/2000), mouse anti-AGR2 (Santa Cruz Biotechnology, Clone 6C5, 1/1000), rabbit anti-XBP1s (Cell Signaling Technologies, clone E9V3E, 1/1000), rabbit anti-BiP (Cell Signaling Technologies, clone C50B12, 1/1000), rabbit β-Tubulin-HRP (Abcam, ab21058, 1/5000). For Avi-tag IP's: rabbit AGR2 (Cell Signaling Technologies, Clone D9V2F, 1/1000), mouse FLAG-M2-HRP (Sigma, A8592, 1/1000), Streptavidin-HRP (CST, 3999S, 1/1000), rabbit β-Tubulin-HRP (Abcam, ab21058, 1/5000). For in vivo IP: mouse AGR2 (Santa Cruz Biotechnology, Clone 6C5, 1/1000), rabbit anti-IRE1β (NY109, gift from David Ron, University of Cambridge, 1/2000), rabbit β-Tubulin-HRP (Abcam, ab21058, 1/5000). For gene silencing: mouse anti-AGR2 (Santa Cruz Biotechnology, Clone 6C5, 1/1000), rabbit anti-XBP1s (Cell Signaling Technologies, clone E9V3E, 1/1000), rabbit β-Tubulin-HRP (Abcam, ab21058, 1/5000). Secondary antibodies were goat anti-rabbit IgG-HRP (Dako, P0448, 1/2000) and goat anti-mouse IgG-HRP (Dako, P0447, 1/1500) for capturing on an Amersham 600 Imager (GE Healthcare). For capturing on the Odyssey imaging system, secondary antibodies were anti-mouse IRDYe 800CW and anti-rabbit IRDye 680RD (Li-Cor).

## RT-qPCR

Total RNA was isolated from cells using the Aurum Total RNA Mini kit (Bio-Rad) according to the manufacturer's instructions. cDNA was synthesized using SensiFast cDNA synthesis kit

(Meridian Bioscience). RT-qPCR was carried out in 384-well plates using SensiFAST SYBR No-ROX kit (Meridian Bioscience) according to the manufacturer's protocol. Preparation of mixes and transferring to 384-well plates was performed using the Janus automatic liquid sample handling station (Perkin-Elmer) or the I.DOT dispenser (Dispendix). All data analyses were performed by qbase+ (CellCarta). For each experiment, the most stable reference genes were selected using qbase+ and are mentioned in each figure or figure legend. The following primer pairs were used for RT-qPCR:

| Gene | Forward | Reverse |
| --- | --- | --- |
| hAGR2 | ACCACAGTCAAACCTGGAGC | AGTTGGTCACCCCAACCTCT |
| hXBP1S | CTGAGTCCGAATCAGGTGCAG | ATCCATGGGGAGATGTTCTGG |
| hXBP1T | TGGCCGGGTCTGCTGAGTCCG | ATCCATGGGGAGATGTTCTGG |
| hBLOC1S1 | ATGCTGTCCCGCCTCCTAA | GGAAGGGGCAGACTGCAGCT |
| hACTB | AGCCACATCGCTCAGACAC | GCCCAATACGACCAAATCC |
| hPPARA | TGACCTTGATTTATTTTGCATACC | CGAGCAAGACGTTCAGTCCT |
| hHPRT1 | CCAACCGCGAGAAGATGA | CCAGAGGCGTACAGGGATAG |
| hERN1 | TTTGGAAGTACCAGCACAGTG | TGCCATCATTAGGATCTGGGA |
| hERN2 | TTCTCTACCTTGGACACCCAGC | GGTCAGTCCACGAGGCACC |
| hUBC | GCACTGGAACTGGATGACAG | TTTAGAAGGCCAGGACGATCT |
| hHSPA5 | GGGAACGTCTGATTGGCGAT | CGTCAAAGACCGTGTTCTCG |

### XBP1 splicing assay

Total RNA was isolated and transcribed to cDNA as described for "RT-qPCR". The XBP1 spliced region was amplified using Q5 high fidelity polymerase (NEB) and primers: fwd 5'-TGAAAAACAG AGTAGCAGCTCAGA-3', rev 5'- TCTGGGTAGACCTCTGGGA GCTCC-3'. PCR reaction was separated on 2.5% agarose gel.

### Quantification of cell death

For flow cytometry, cells were seeded in six-well plates at ~80% confluency. The next day, IRE1β expression was induced using 1 µg/ml doxycycline, supplemented with either 1 µM 4µ8C or DMSO. Forty-eight to seventy-two hours later, the supernatant was collected, cells were washed in PBS which was added to the supernatant, and cells were dissociated with Accutase (Biolegend) and added to the supernatant. Cells were spun down and stained 20' at 4DC with eBioscience™ Fixable Viability Dye eFluor™ 780 (Invitrogen), washed and stained 15' at RT with Annexin V PE (BD Pharmingen™) in annexin binding buffer. Cells were diluted in binding buffer and measured on a BD LSRII. Flow cytometry data was processed using FlowJo X.

### Gel filtration oligomerization assays

For Calu-1 cells, IRE1β expression was induced with 1 µg/ml doxycycline for 16 h. A 145-cm² culture dish at 80% confluency was lysed in 600 µl lysis buffer buffer (150 mM NaCl, 25 mM Tris pH 8.0, 20 mM dodecylmaltoside, 5 mM β-mercaptoethanol with cOmplete mini protease inhibitor cocktail (Roche) and PhosSTOP (Roche)) at 4 °C for 1 h under continuous agitation. For assays in HEK293T cells, cells were seeded at $5.10^6$ cells/60-cm² dish and

transfected with pDEST51-IRE1β-msfGFP plasmid alone or a 4.3:1 and 2:1 molar ratio of pDEST51-AGR2 and pDEST51-IRE1β-msfGFP plasmid (6.6 µg in total) via Calcium-Phosphate transfection. Each dish was collected and lysed identically to the Calu-1 cells above. Lysates were cleared by centrifugation at 4 °C, $16.000× g$ for 20 min. Fractionation of lysates was carried out on a Superose6 Increase 10/300 GL (Cytiva) size exclusion chromatography column equilibrated in running buffer (150 mM NaCl, 25 mM Tris pH 8.0, 0.5 mM dodecylmaltoside, 5 mM β-mercaptoethanol) at 0.5 ml/min. Detection of fluorescent IRE1β-msfGFP was carried out on an RX-20Axs fluorescence detector (Shimazu, Ex: 485 nm/Em: 510 nm) in line with the UV detector. For IRE1β-FLAG detection, 0.25 ml fractions were collected and assayed by SDS-PAGE (Criterion TGX 4–20% (Bio-Rad)) and western blot using HRP-coupled anti-FLAG-M2 (Sigma, A8592) and anti-AGR2 (Cell Signaling Technologies, Clone D9V2F, 1/1000).

### Gene silencing

AGR2 knockdown in LS174T cells was obtained by reverse transfection of siRNA (siGENOME, Horizon Discovery, D-003626-01; D-003626-03; D-001206-14) using DharmaFECT1 transfection reagent (Horizon Discovery) according to the manufacturer's protocol. After 24 h, culture medium was replaced and supplemented with 1 µM 4µ8C or DMSO. Forty-eight hours after transfection, cells were harvested. Total RNA was isolated, transcribed to cDNA and RT-qPCR was performed as described for "RT-qPCR" above. To assay AGR2 protein levels, cells were resuspended in lysis buffer (1% NP-40, 10% glycerol, 250 mM NaCl, 20 mM HEPES pH 7.9, 1 mM EDTA, 2 mM DTT with cOmplete mini protease inhibitor cocktail (Roche) and PhosSTOP (Roche)) at 4 °C for 15 min. Lysates were cleared by centrifugation at 4 °C, $16.000× g$ for 20 min. Total protein was measured using Bradford assay (Bio-Rad).

### Image quantification

Western blot band intensities were quantified using ImageJ (Schneider et al, 2012) and normalized over input or overloading control (normalization strategy is indicated in each figure legend).

### Structural modeling

All structure prediction models were generated using the Alphafold2.3.1 software (Jumper et al, 2021) on an HPC cluster (Vlaams Supercomputer Centrum, Belgium). For each job, five different seed models were requested that each were used to generate 5 models, giving rise to a total of 25 different models. All predictions were scored and ranked based on their PTM values. Figures were visualized using PyMOL (Schrodinger, 2015).

## Data availability

Physical materials established in this manuscript can be requested through the corresponding author. The datasets produced in this study are available in the following database: Protein interaction AP-MS data: ProteomeXchange Consortium via the PRIDE partner repository with the dataset identifier PXD042756.

## Peer review information

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

## Acknowledgements

SJ, SE, and SNS are supported by the Vlaams Instituut voor Biotechnologie (VIB). This research was further supported by Fonds Wetenschappelijk Onderzoek (FWO) (1228923N to EC, G017521N to SJ, and G049820N, G0H1222N, S000722N, S002322N to SNS), an ERC Consolidator grant (DCRIDDLE- 819314 to SJ) and the Geconcerteerde Onderzoeksacties, Special Research fund, Ghent University (BOF16/GOA/023 to SE). LN is supported by

Medical Research Council DTP and Gates Cambridge PhD programme. We would like to thank the VIB Proteomics core, the VIB Flow core and the VIB-IRC Tissue Culture Core for training, support and access to the instrument park and facility. We thank David Ron for feedback on the manuscript, and for providing IRE1β antiserum. We would like to thank Michael J Grey for constructive discussions and advice.

## Author contributions

**Eva Cloots**: Conceptualization; Data curation; Formal analysis; Funding acquisition; Validation; Investigation; Writing—original draft; Writing—review and editing. **Phaedra Guilbert**: Formal analysis; Investigation; Writing—review and editing. **Mathias Provost**: Resources; Investigation; Methodology; Writing—review and editing. **Lisa Neidhardt**: Resources; Writing—review and editing. **Evelien Van De Velde**: Investigation. **Farzaneh Fayazpour**: Investigation. **Delphine De Sutter**: Investigation. **Savvas Savvides**: Resources; Investigation; Methodology; Writing—review and editing. **Sven Eyckerman**: Conceptualization; Supervision; Funding acquisition; Methodology; Writing—review and editing. **Sophie Janssens**: Conceptualization; Supervision; Methodology; Writing—original draft; Writing—review and editing.

## Disclosure and competing interests statement

The authors declare no competing interests.

# Expanded View Figures

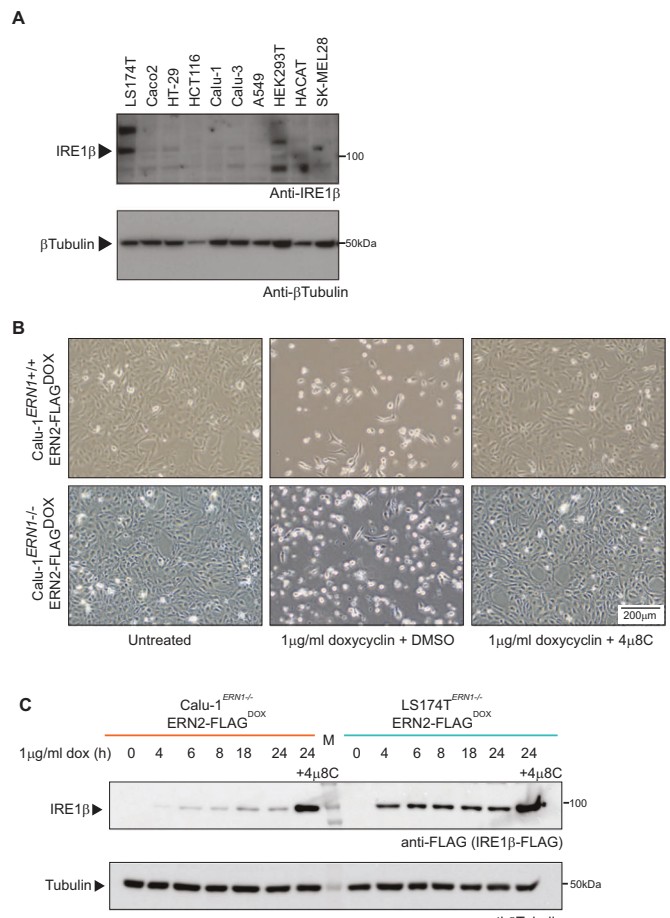

**Figure EV1. Validation of LS174T^ERN1-/-IRE1βFLAG-DOX and Calu-1 ^ERN1-/-IRE1βFLAG-DOX model systems.**

(**A**) IRE1β expression in cell lines. Proteins were extracted and probed for IRE1β expression via immunoblot. Tubulin was used as a loading control. (**B**) Photographs showing the phenotype of cultures overexpressing IRE1β-FLAG in IRE1α wild-type (*ERN1^+/+*) and IRE1α deficient (*ERN1^−/−*) cells. Images of IRE1α deficient (*ERN1^−/_*) cultures are the same images as shown in Fig. 1C. Left panels show untreated cultures, middle panels show cultures treated with 1 μg/ml doxycycline for 72 hours, right panels show cultures treated with both 1 μg/ml doxycycline and 1 μM IRE1 endonuclease inhibitor 4μ8C. Scale bar represents 200 μm. (**C**) Quantification of IRE1β transgene expression over time in Calu-1^ERN1-/-IRE1βFLAG-DOX and LS174T^ERN1-/-IRE1βFLAG-DOX cells by western blot. Cultures were treated with 1 μg/ml doxycycline for the indicated times, protein lysates were probed for IRE1β-FLAG expression and tubulin as a loading control.

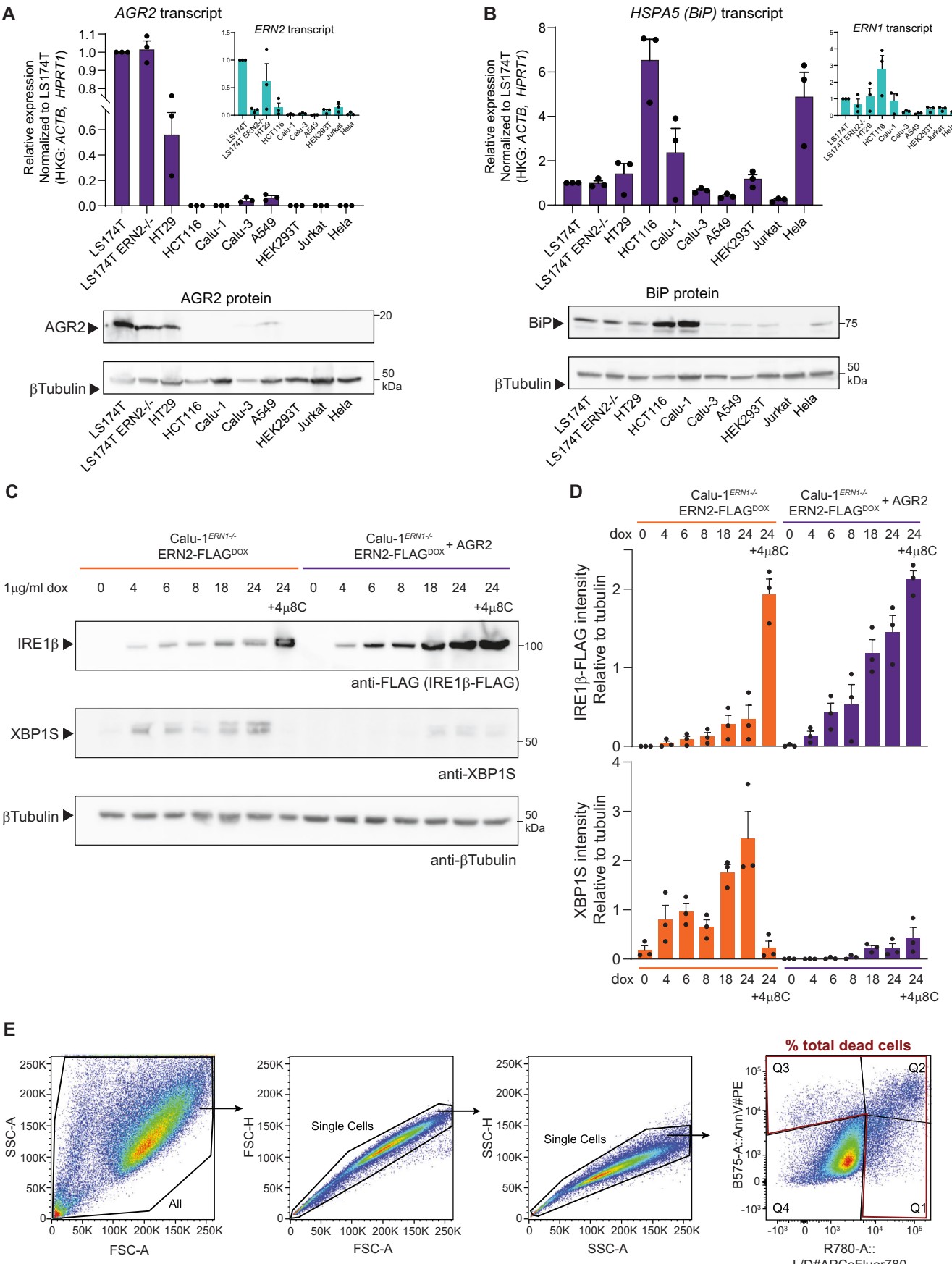

◀ **Figure EV2. AGR2 expression is restricted to colon epithelial cell lines and affects Calu-1[ERN1-/-IRE1βFLAG-DOX] phenotype.**

(A) *AGR2* (purple) and *ERN2* (blue insert, same data as shown in Fig. 1A) transcript expression in cell lines assayed by RT-qPCR. $N = 3$ culture dishes were sampled and *AGR2* and *ERN2* expression is shown relative to the expression detected in LS174T parental cells. Error bars show SEM. Bottom picture shows protein expression by western blot. Protein lysates were probed for AGR2 and tubulin was used as a loading control. (B) *HSPA5* (purple) and *ERN1* (blue insert, same data as shown in Fig. 1A) transcript expression in cell lines assayed by RT-qPCR. $N = 3$ culture dishes were sampled and *HSPA5* and *ERN2* expression is shown relative to the expression detected in LS174T parental cells. Error bars show SEM. Bottom picture shows protein expression by western blot. Protein lysates were probed for BiP and tubulin was used as a loading control. (C) IRE1β-FLAG transgene expression over time by western blot in cell lysates derived from Calu-1[ERN1-/-IRE1βFLAG-DOX] co-expressing ER-targeted BirA as a control protein (left, orange), or AGR2 (right, purple). Cells received 1 μg/ml doxycycline to induce expression of IRE1β. Protein lysates were prepared at the indicated times and probed for IRE1β-FLAG expression, XBP1S and tubulin as a loading control. (D) Quantification of IRE1β-FLAG and XPP1S protein levels normalized to tubulin in three replicate experiments represented in (C). Error bars represent SEM. (E) Gating strategy to assess cell death. Doublets were gated out and dead cells were gated on via Annexin V and Live/Dead positive staining. All cells staining positive for a single, or both cell death markers were considered dead (red gate).

			

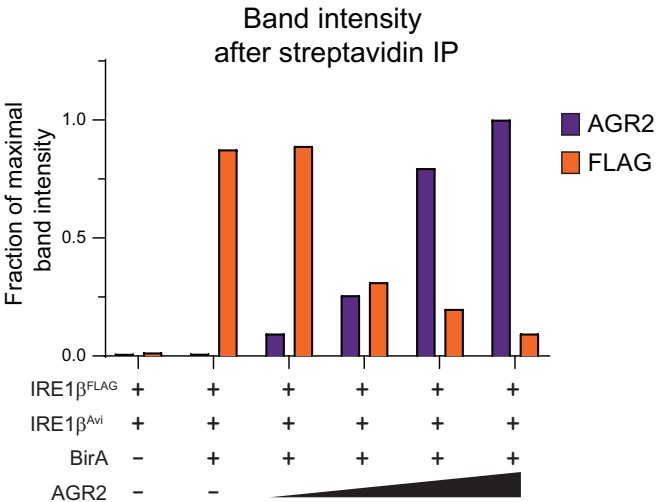

**Figure EV3.  Inverse relationship between AGR2 IP and IRE1β complex formation.**

Quantification of immunoblots in Fig. 4F ($n = 2$). IRE1β-FLAG band intensity was normalized to input expression levels. Band intensity values are expressed as a fraction of the maximal intensity that was obtained in each respective experiment.

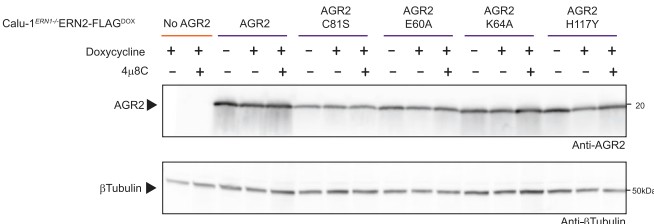

**Figure EV4. AGR2 expression in stably transduced cell lines.**

AGR2 expression in cultures analyzed in Fig. 5D,E. Tubulin was used as a loading control.

**A**

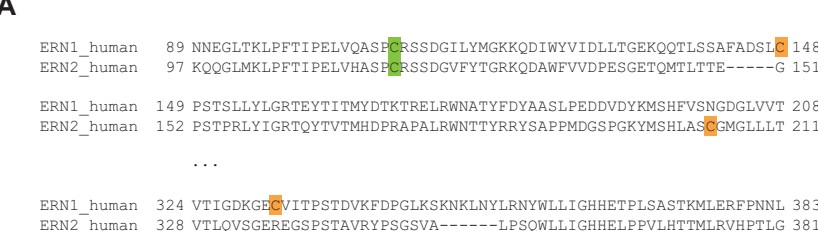

```
ERN1_human   89 NNEGLTKLPFTIPELVQASPCRSSDGILYMGKKQDIWYVIDLLTGEKQQTLSSAFADSLC 148
ERN2_human   97 KQQGLMKLPFTIPELVHASFCRSSDGVFYTGRKQDAWFVVDPESGETQMTLTTE-----G 151

ERN1_human  149 PSTSLLYLGRTEYTITMYDTKTRELRWNATYFDYAASLPEDDVDYKMSHFVSNGDGLVVT 208
ERN2_human  152 PSTPRLYIGRTQYTVTMHDPRAPALRWNTTYRRYSAPPMDGSPGKYMSHLASCGMGLLLT 211

                    . . .

ERN1_human  324 VTIGDKGECVITPSTDVKFDPGLKSKNKLNYLRNYWLLIGHHETPLSASTKMLERFPNNL 383
ERN2_human  328 VTLQVSGEREGSPSTAVRYPSGSVA------LPSQWLLIGHHELPPVLHTTMLRVHPTLG 381
```

**B**

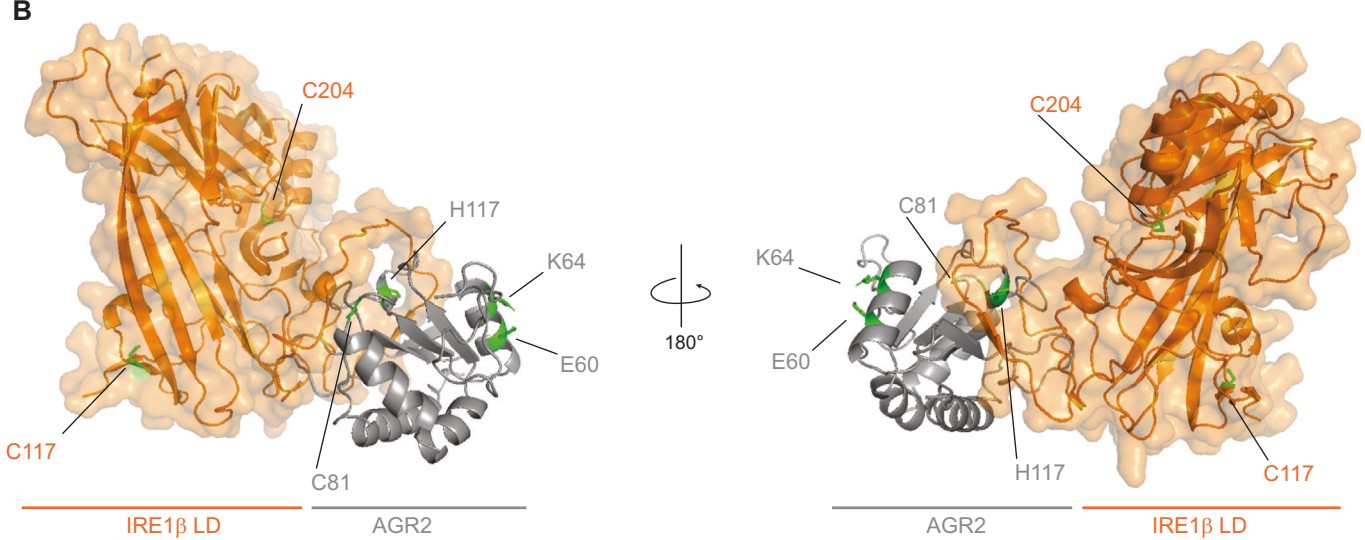

**C**

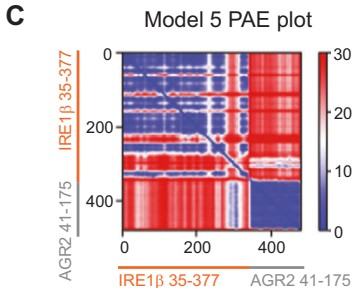

Model 5 PAE plot

Figure EV5.  **AGR2 is predicted to bind a flexible loop region in the IRE1β luminal domain.**

(**A**) BLAST alignment of the regions containing cysteines in human IRE1α and IRE1β. Green square indicates the sole conserved cysteine in IRE1α and IRE1β luminal domain, orange squares show cysteines present in only one of the paralogues. (**B**) Highest scoring AlphaFold2-Multimer model (pTM score = 0.662), modeled using IRE1β residues 35–377 (Uniprot Q76MJ5) and AGR2 residues 41–175 (Uniprot O95994). The IRE1β luminal domain is shown in orange and AGR2 in grey. Labels indicate the highlighted green residues. (**C**) Predicted aligned error (PAE) plot for the model shown in B.

