## [Peer Review File · The EMBO Journal]

Activation of goblet-cell stress sensor IRE1 β is controlled by the mucin chaperone AGR2

Eva Cloots, Phaedra Guilbert, Mathias Provost, Lisa Neidhardt, Evelien Van De Velde, Farzaneh Fayazpour, Delphine De Sutter, Savvas Savvides, Sven Eyckerman, and Sophie Janssens

DOI: [10.15252/emj.2023114710](https://doi.org/10.15252/emj.2023114710)

Corresponding author(s): *Sophie Janssens (Sophie.Janssens@irc.vib-ugent.be)*

Review Timeline:

Submission Date:	14th Jun 23
Editorial Decision:	30th Jun 23
Appeal Received:	3rd Jul 23
Editorial Decision:	1st Aug 23
Revision Received:	16th Oct 23
Editorial Decision:	3rd Nov 23
Revision Received:	7th Nov 23
Editorial Decision:	10th Nov 23
Revision Received:	13th Nov 23
Accepted:	15th Nov 23

Editor: *William Teale*

Transaction Report:

Dear Sophie,

Thank you for submitting your manuscript entitled 'Activation of goblet cell stress sensor IRE1 β is controlled by the mucin chaperone AGR2' (EMBOJ-2023-114710) to our editorial office. We have now considered the study within our editorial team, and unfortunately come to the conclusion that we cannot offer publication in The EMBO Journal. We realize that your identification of AGR2 as a direct ER-luminal interactor and functional gate of IRE1 β is likely to be of interest to the field. Furthermore, the experiments you use to test the role of the IRE1 β -AGR2 interaction in a disease context are certainly intriguing. However, we are not convinced that your manuscript provides a sufficiently compelling conceptual advance to make the study a strong candidate for publication in a broad general journal like The EMBO Journal at this stage.

However, given the potential value of your results to researchers directly in the field, we feel that the study should be well-suited for our sister journal EMBO Reports. I have therefore briefly discussed the work with the editors there, who would indeed be pleased to send your work for in-depth external review, without need for prior reformatting. Should you be interested in this option, please simply follow the link below for transfer. Martina Rembold, Senior Editor at EMBO Reports (m.rembold@emboreports.org), will be happy to answer any questions you may have.

I am sorry that I cannot be more positive for The EMBO Journal on this occasion, but very much hope that you will find this transfer option worthwhile.

Yours sincerely,

William

William Teale, PhD
Editor
The EMBO Journal
w.teale@embojournal.org

** As a service to authors, EMBO Press provides authors with the possibility to transfer a manuscript that one journal cannot offer to publish to another EMBO publication or the open access journal Life Science Alliance launched in partnership between EMBO Press, Rockefeller University Press and Cold Spring Harbor Laboratory Press. The full manuscript and if applicable, reviewers' reports, are automatically sent to the receiving journal to allow for fast handling and a prompt decision on your manuscript. For more details of this service, and to transfer your manuscript please click on Link Not Available. **

Appeal received on 3rd July 2023.

Dear Sophie,

Thank you again for the submission of your manuscript entitled "Activation of goblet cell stress sensor IRE1 β is controlled by the mucin chaperone AGR2" and for your patience (and trust in our editorial team) during the review process. We have now received reports from all three referees, which I copy below.

As you can see from their comments, all referees were fully supportive of publication in The EMBO Journal. Referee 1 suggests some strengthening experiments; please consider these suggestions and respond in a document that addresses all points raised.

Based on the overall interest expressed in the reports, therefore, I would like to invite you to address the comments of all referees in a revised version of the manuscript. I should add that it is The EMBO Journal policy to allow only a single major round of revision and that it is therefore important to resolve the main concerns at this stage. I believe the concerns of the referees are reasonable and addressable, but please contact me if you have any questions, need further input on the referee comments or if you anticipate any problems in addressing any of their points. I am available to Zoom call if you would like to discuss Referee 1's report. Please, follow the instructions below when preparing your manuscript for resubmission.

I would also like to point out that as a matter of policy, competing manuscripts published during this period will not be taken into consideration in our assessment of the novelty presented by your study ("scooping" protection). Please contact me if you see a paper with related content published elsewhere to discuss the appropriate course of action.

Again, please contact me at any time during revision if you need any help or have further questions.

Thank you very much again for the opportunity to consider your work for publication. I look forward to your revision.

Best regards,

William

William Teale, Ph.D.
Editor
The EMBO Journal

When submitting your revised manuscript, please carefully review the instructions below and include the following items:

- 1) a .docx formatted version of the manuscript text (including legends for main figures, EV figures and tables). Please make sure that the changes are highlighted to be clearly visible.
- 2) individual production quality figure files as .eps, .tif, .jpg (one file per figure).
- 3) a .docx formatted letter INCLUDING the reviewers' reports and your detailed point-by-point response to their comments. As part of the EMBO Press transparent editorial process, the point-by-point response is part of the Review Process File (RPF), which will be published alongside your paper.
- 4) a complete author checklist, which you can download from our author guidelines ([https://wol-prod-cdn.literatumonline.com/pb-assets/embo-site/Author Checklist%20-%20EMBO%20J-1561436015657.xlsx](https://wol-prod-cdn.literatumonline.com/pb-assets/embo-site/Author%20Checklist%20-%20EMBO%20J-1561436015657.xlsx)). Please insert information in the checklist that is also reflected in the manuscript. The completed author checklist will also be part of the RPF.
- 5) Please note that all corresponding authors are required to supply an ORCID ID for their name upon submission of a revised manuscript.
- 6) We require a 'Data Availability' section after the Materials and Methods. Before submitting your revision, primary datasets produced in this study need to be deposited in an appropriate public database, and the accession numbers and database listed under 'Data Availability'. Please remember to provide a reviewer password if the datasets are not yet public (see <https://www.embopress.org/page/journal/14602075/authorguide#data deposition>). If no data deposition in external databases is needed for this paper, please then state in this section: This study includes no data deposited in external repositories. Note that

the Data Availability Section is restricted to new primary data that are part of this study.

Note - All links should resolve to a page where the data can be accessed.

8) For data quantification: please specify the name of the statistical test used to generate error bars and P values, the number (n) of independent experiments (specify technical or biological replicates) underlying each data point and the test used to calculate p-values in each figure legend. The figure legends should contain a basic description of n, P and the test applied. Graphs must include a description of the bars and the error bars (s.d., s.e.m.).

9) We would also encourage you to include the source data for figure panels that show essential data. Numerical data can be provided as individual .xls or .csv files (including a tab describing the data). For 'blots' or microscopy, uncropped images should be submitted (using a zip archive or a single pdf per main figure if multiple images need to be supplied for one panel). Additional information on source data and instruction on how to label the files are available at .

10) We replaced Supplementary Information with Expanded View (EV) Figures and Tables that are collapsible/expandable online (see examples in <https://www.embopress.org/doi/10.15252/embj.201695874>). A maximum of 5 EV Figures can be typeset. EV Figures should be cited as 'Figure EV1, Figure EV2" etc. in the text and their respective legends should be included in the main text after the legends of regular figures.

12) Our journal encourages inclusion of *data citations in the reference list* to directly cite datasets that were re-used and obtained from public databases. Data citations in the article text are distinct from normal bibliographical citations and should directly link to the database records from which the data can be accessed. In the main text, data citations are formatted as follows: "Data ref: Smith et al, 2001" or "Data ref: NCBI Sequence Read Archive PRJNA342805, 2017". In the Reference list, data citations must be labeled with "[DATASET]". A data reference must provide the database name, accession number/identifiers and a resolvable link to the landing page from which the data can be accessed at the end of the reference. Further instructions are available at .

Further instructions for preparing your revised manuscript:

We realize that it is difficult to revise to a specific deadline. In the interest of protecting the conceptual advance provided by the work, we recommend a revision within 3 months (30th Oct 2023). Please discuss the revision progress ahead of this time with the editor if you require more time to complete the revisions. Use the link below to submit your revision:

Referee #1:

This study demonstrates the mechanism of activation of IRE1beta in a cell line expressing endogenous IRE1beta. The authors show that the chaperone ARG2 binds to the IRE1beta ER-sensor, resulting in the inactivation of the protein. AGR2 deficiency elicits constitutive activation of IRE1beta. This study and the accompanying paper convincingly support the model that AGR2 is a master regulator of IRE1beta. This is a fundamental discovery of crucial importance in the cell homeostasis maintenance field as it provides a clear conceptual advance.

The study's strength is the use of cells expressing physiologically the IRE1beta machinery to demonstrate AGR2 involvement in IRE1beta activation. The authors also provide a possible mechanism for the loss of intestinal homeostasis observed in patients carrying the AGR2 H117Y mutation. Some of the comments below may have been addressed or are related to the submitted accompanying paper.

- 1) The title in the paragraph related to Figure 4 (p 9), mentioning that ARG2 disrupts IRE1 beta dimers, is not fully supported by the data that primarily rely on gel filtration in whole cell lysate. While consistent with the literature, it is difficult to exclude binding to other proteins.
- 2) In Figure 4F, it is not clear what the ratio representing ARG2 expression in the figure means. Are these indicative of DNA concentration or protein expression? How is that related to molar concentrations, as indicated in the text? This should be clarified.
- 3) In Figure 4, The interaction with BiP would be of interest to assess in this figure as Bip was identified in the mass-spec pull-down as binding both IRE1alpha and beta. Along the same line, assessing BiP expression in the Supp2A figure along with AGR2 expression would be very informative. I wondered if low BiP expression in Goblet and LS174T cells could explain the evolution of an AGR2 sensor.
- 4) Related to the point above, the expression of IRE1alpha in the various cell lines tested in Figure 1A would be informative. IRE1alpha is often seen as a housekeeping gene that lacks specific regulation.
- 5) In the accompanying paper, it is shown that MUC2 expression releases AGR2 inhibition of IRE1beta. Here the authors have a cellular model that best recapitulated IRE1beta physiological context. It would be complementary to the other study to use their dox inducible system to express AGR2 substrate and see if, in their model, they recapitulate IRE1 beta activation.

Minor point

Figure 4E, the band migrating below IRE1 in the input should be labeled as non-specific if this is the case.

Referee #2:

This manuscript by Cloots et al employs a cell biological approach to identify AGR2 as a goblet-cell specific protein involved in the selective regulation of IRE1beta, but not IRE1alpha. Using mammalian cell culture models that express IRE1beta at either high (goblet cell derived LS174T cells) or low levels (Calu cells), they depleted endogenous IRE1alpha and then expressed dox-inducible IRE1beta. They found that cells lacking endogenous IRE1beta showed higher basal levels of toxicity and IRE1 activity in response to dox-induced expression of IRE1beta, as compared to cells expressing endogenous IRE1beta. This suggested that cells that endogenously express IRE1beta (e.g., goblet cells) must also express some other factor that selectively represses IRE1beta activity. Using a proteomic-based approach to study interactomes of IRE1alpha or IRE1beta, they identified AGR2 as a specific protein that co-purifies with IRE1beta goblet-derived LS174T cells. They go on to show that overexpression of AGR2 can suppress toxicity and IRE1beta activity in IRE1alpha-depleted Calu cells. They go on to demonstrate that AGR2 appears to disrupt IRE1beta dimers, providing a potential mechanism to explain the repression of IRE1beta activity observed for this protein. Finally, the authors show that catalytically inactive AGR2 (where a Cys in the thioredoxin active site is disrupted) and a disease-associated mutation (H117Y) impaired the ability to disrupt IRE1beta dimers and mitigate toxicity in Calu cells expressing IRE1beta. Collectively, these results identify AGR2 as a specific repressor of IRE1beta activity in specialized cells such as goblet cells.

This cell biological manuscript, in combination with the co-submitted manuscript from Niedhardt et al that applies a more biochemical approach, clearly demonstrates an important role for AGR2 in suppressing IRE1beta activity. These two manuscripts, which approach the problems in complementary ways, provide new mechanistic insight into this problem and highlight unique relationships between cell-type specific regulation of UPR stress sensors. The experiments in this manuscript are well performed and interpreted, all of which support the conclusions highlighted in the text. I do have a few minor comments that could be addressed in revision, but these two manuscripts, especially when viewed together, are quite strong in demonstrating an important role for AGR2 in regulating IRE1beta in specific cell types such as goblet cells.

COMMENTS.

1. The RIDD results in Figure 3A. I think it would be useful to repeat this a few more times to get a better idea on how AGR2 regulates RIDD in the Calu cells. The spread on the BLOC1S1 is quite large. I get the point that this reflects RNase activity and the XBP1s/t data is there too, but for completeness improving that figure a bit would be helpful.
2. The authors show that catalytically inactive and disease-associated AGR2 mutants impair the ability for AGR2 to inhibit IRE1beta dimerization and IRE1beta-associated cellular toxicity. It would be good to also include evidence for impaired IRE1beta RNase repression in these experiments (monitoring RIDD targets and XBP1s).
3. Could overexpression of mucin 'de-repress' the reduced IRE1beta activity observed in Calu cells expressing both AGR2 and dox-inducible IRE1beta? I realize that a similar experiment was performed in the accompanying paper, but it would be good to show similar results in this manuscript as well to provide some evidence to the idea that AGR2 specifically senses mucin ER protein homeostasis. This is alluded to in the text and adding an experiment to address that would be nice to help paint the full picture in this specific manuscript.

Referee #3:

Accumulating misfolded proteins in the endoplasmic reticulum (ER) of eukaryotic cells trigger the so-called unfolded protein response (UPR), a transcriptional and translational response that activates the expression of molecular chaperones to cope with the misfolded proteins in the ER and down regulates general translation to reduce the folding load of incoming newly synthesized proteins. One of the central regulators of the UPR is Ire1, a kinase and endonuclease that activates the transcription factor XBP-1. In mammalian cells there are two isoforms of Ire1, the ubiquitously expressed Ire1 α and Ire1 β , which is only expressed in mucus producing cells like the goblet cells of the intestine. Whereas the regulation of Ire1 α is intensively investigated, regulation of Ire1 β was still poorly studied. Ectopic expression of Ire1 β in many generally used cell lines was shown to lead to growth arrest.

The authors of this study investigated the expression, activation and attenuation of Ire1 β in a cell line derived from goblet cells, LS174T, and, for comparison, the lung epithelial cell line Calu-1 that normally do not express Ire1 β . They confirmed that high level expression of Ire1 β exerts toxicity in Calu-1 cells but not in LS174T cells. They show that expression of Ire1 β induces splicing of XBP-1 mRNA and endonucleolytic degradation of mRNAs and that inhibition of these activities prevents the toxicity effect of Ire1 β expression, suggesting that there is a repressor present in LS174T cells that is absent in other cell lines tested. The authors immunoprecipitated Ire1 β and for comparison Ire1 α from LS174T cells and analyze the co-precipitating proteins by quantitative mass spectrometry. Among the most significantly enriched proteins that specifically co-precipitate with Ire1 β but not Ire1 α was AGR2, a chaperone and protein disulfide isomerase that was already linked to diseases caused by deficient mucus production of goblet cells. The authors demonstrate that Ire1 β is dimeric or oligomeric in vivo and that AGR2 interacts with Ire1 β in vivo and in vitro. They provide evidence that AGR2's interaction leads to monomerization of Ire1 β in cells. Co-expression of AGR2 with Ire1 β in Calu-1 cells alleviates the toxicity effect of Ire1 β overexpression. Down-regulation of AGR2 by siRNA in LS174T cells derepresses Ire1 β activity. Amino acid replacements, including a patient mutation, that compromises the

interaction of AGR2 with Ire1 β leads to de-repression of Ire1 β and increased XBP-1 splicing activity. This is a very interesting study that is complementary to the parallel submitted study by Ron and coworkers (Neidhardt et al). There are a few points the authors should address before publication.

Major criticism

I assume that the supplemental Figures have been named extended view figures in the text and figure legends. A large body of evidence in the manuscript resides on immune blot analysis (often called quantification of protein levels by the authors in the figure legends). However, there is not a single quantitative analysis of the immune blots shown and no statistical significance of differences has been established. This is essential and has to be performed. Although, differences seem obvious in some of the blots (e.g. fig. 1D), in others (e.g. fig. 5c, or XBP1s in fig. S2B) it is not so clear how reproducible such blots were and whether similar changes were also found in repeated experiments. Co-precipitations should be normalized to the amount of the bait protein. If less bait protein is pulled down, it is expected that less prey protein is detected in the immune blot.

Minor comments:

Fig. 3E: the authors show relative expression of AGR2 after siRNA mediated knock-down. However, neither out of the figure nor of the figure legend it becomes clear whether this panel shows qPCR data quantification of the mRNA or immunoblot quantification of the protein levels. This should be stated clearly.

Fig. 3F the authors should clearly state how the changes were determined.

Fig. 6: since this figure has only a single panel the panel letter "A" is unnecessary and confusing. The same applies to Fig. S3/ Fig. EV3.

Fig. EV4: the figure numbering is not shown on the figure.

Referee #1:

This study demonstrates the mechanism of activation of IRE1beta in a cell line expressing endogenous IRE1beta. The authors show that the chaperone ARG2 binds to the IRE1beta ER-sensor, resulting in the inactivation of the protein. ARG2 deficiency elicits constitutive activation of IRE1beta. This study and the accompanying paper convincingly support the model that ARG2 is a master regulator of IRE1beta. This is a fundamental discovery of crucial importance in the cell homeostasis maintenance field as it provides a clear conceptual advance.

The study's strength is the use of cells expressing physiologically the IRE1beta machinery to demonstrate ARG2 involvement in IRE1beta activation. The authors also provide a possible mechanism for the loss of intestinal homeostasis observed in patients carrying the ARG2 H117Y mutation. Some of the comments below may have been addressed or are related to the submitted accompanying paper.

1) The title in the paragraph related to Figure 4 (p 9), mentioning that ARG2 disrupts IRE1 beta dimers, is not fully supported by the data that primarily rely on gel filtration in whole cell lysate. While consistent with the literature, it is difficult to exclude binding to other proteins.

We agree with the reviewer that we cannot draw this conclusion with 100% certainty based on this sole gel filtration experiment and can only state that ARG2 drives IRE1 β toward a monomeric state. However, as the reviewer correctly mentions, this conclusion is also based on literature, and on the data in the accompanying paper where gel filtrations were performed with recombinant proteins. Still, we addressed this editorially by replacing the word dimer by oligomer in titles and the abstract, and by explicitly referring to the data by Neidhardt et al in the final conclusion sentence of the paragraph.

This paragraph has also been rewritten to more clearly/concisely acknowledge the caveats of whole cell extract gel filtration (see yellow highlights in manuscript).

2) In Figure 4F, it is not clear what the ratio representing ARG2 expression in the figure means. Are these indicative of DNA concentration or protein expression? How is that related to molar concentrations, as indicated in the text? This should be clarified.

The ratio is indicative of the molar concentration of specifically the cDNA fragment in the plasmid. We understand that this is confusing, and the explicit mentioning of the ratio's is in hindsight not that informative, since we cannot be confident that this translates to similar protein ratios. The figure will be adapted and the ratios will be removed (only the increasing black bar will remain), and the definition of the used plasmid ratios will be explained in more detail in the methods section.

3) In Figure 4, The interaction with BiP would be of interest to assess in this figure as BiP was identified in the mass-spec pull-down as binding both IRE1alpha and beta. Along the same line, assessing BiP expression in the Supp2A figure along with ARG2 expression would be very informative. I wondered if low BiP expression in Goblet and LS174T cells could explain the evolution of an ARG2 sensor.

It is an interesting thought that ARG2 could have evolved as an alternative for BiP in UPR regulation. However, when looking into the single cell RNA-Seq dataset published by Haber

et al (2017, Nature), it appears that goblet cells show very high expression levels of both BiP and AGR2 (Fig R1).

Figure R1. Expression of BiP (Hspa5) and Agr2 in mouse small intestine as assessed by scRNA-Seq (Haber et al. Nature. 2017, accessed through https://singlecell.broadinstitute.org/single_cell)

We also assessed BiP expression in the cell lines from Fig. EV2A (now added as panel EV2B) and confirm that the variation in expression levels found in vivo (Fig R1) is also found in epithelial cell lines (Fig EV2A, B in updated manuscript). The BiP expression level in LS174T cells is comparable to 293T and Calu-3 cells and significantly lower than HCT116 and Hela cells. Of note, while HSPA5/BiP expression appears to match IRE1 α /ERN1 expression, AGR2 expression appears to match IRE1 β /ERN2 expression (Fig EV2)

We tried to address the interaction of BiP with IRE1 α and IRE1 β to check for potential specificity in BiP binding. In an early study by the David Ron Lab (Bertolotti et al, 2000, Nat Cell Biol), an interaction between IRE1 β and BiP has been observed, though in a more recent study the interaction with IRE1 β appeared noticeably weaker when directly comparing IRE1 α -BiP with IRE1 β -BiP interactions (Oikawa et al, 2012, Plos One). Recently, a protocol was published by the Ron lab that decreases non-specific binding of BiP during IP conditions (Amin-Wetzel et al, 2019, eLife), but BiP co-IP experiments remain tricky to perform and interpret. When we performed this new, optimized protocol, the results of the different experiments were somewhat variable which complicates the interpretation. One experiment in which we used a Cell Signaling antibody for BiP demonstrated a clear reduction of IRE1 β -BiP complexes compared to IRE1 α -BiP, with a clear ER stress-dependent reduction in the IRE1 α -BiP complex as expected (Fig R2A). As the BiP signal we obtained was rather weak and required long exposure times, we received a more sensitive anti-BiP antibody from David Ron and performed additional experiments. To our surprise, with his antibody, we obtained different results and now observed a strong interaction between IRE1 β and BiP. In contrast to our expectations, we did no longer observe the ER stress dependent release of BiP upon thapsigargin treatment, which was obvious in earlier experiments (Fig R2B). Considering these results, we do not think that these IP experiments are sufficiently convincing and several additional experiments utilizing other technologies and model systems would be required to make definitive claims about differences in strength between IRE1 α versus IRE1 β binding to BiP. Since this would bring us away from the focus of the current manuscript (e.g. the ability of AGR2 to tune IRE1 β activity), we would prefer to leave this question open for now. We do acknowledge that the precise role of BiP in IRE1 β regulation is a relevant question in the bigger picture of IRE1 β regulation. In the discussion, we now acknowledge that this remains a somewhat open question, but we also explicitly refer to the recombinant data in the manuscript by Neidhardt et al to support the notion that

BiP is not strictly required for AGR2 to inhibit IRE1β dimerization, even though there appears to be a certain degree of binding by BiP to IRE1β in cellular systems. Additionally, Neidhardt et al were able to demonstrate a lack of response of IRE1β to BiP depletion, also arguing against a primary role for BiP in IRE1β regulation. In more speculative parts of our discussion on whether AGR2 and BiP provide unique mechanisms for IRE1β and IRE1α activation respectively, we have adapted sentences that imply a strong specificity in binding to instead refer to a specificity in regulatory activity.

A Exp 1 – BiP stained with Cell Signaling Tech Ab

B Exp 2 – BiP stained with David Ron Ab

Figure R2. Co-immunoprecipitation of BiP with IRE1 proteins. Co-IP was performed according to the protocol by Amin-Wetzel et al (2019, eLife) using IRE1-Avi as bait proteins, and immunoblotting for BiP to assess co-precipitation.

4) Related to the point above, the expression of IRE1 alpha in the various cell lines tested in Figure 1A would be informative. IRE1 alpha is often seen as a housekeeping gene that lacks specific regulation.

RT-qPCR data for IRE1α expression in the cell lines was added (see response to comment above) - overall, ERN1 transcript could be readily detected in all cell lines tested with varying expression levels, supporting the lack of cell-specific restriction compared to IRE1β.

5) In the accompanying paper, it is shown that MUC2 expression releases AGR2 inhibition of IRE1beta. Here the authors have a cellular model that best recapitulated IRE1beta physiological context. It would be complementary to the other study to use their dox inducible system to express AGR2 substrate and see if, in their model, they recapitulate IRE1 beta activation.

This is a highly relevant question, but unfortunately highly challenging to address in our cell models for the following reasons:

- The Invitrogen ViraPower TRex dox-inducible system already takes up around 5kb of the viral genome, meaning only smaller genes of interest can be incorporated. We have already found how challenging it is to prepare virus carrying IRE1α or IRE1β using this system (which are under 3kb in size), and the largest MUC2 fragment tested in the accompanying paper is ~4.2kb in size. Obtaining sufficient titers and a validated inducible system was not feasible in the normal timeframe of a revision.

- To investigate this point in a shorter timeframe, we tested transient transfection, but LS174T cells turn out to be incredibly difficult to transfect. Lipofectamine 3000 and LTX yield

transfection rates of maximum 3-5% (Figure R3), and Lonza nucleofection gave very variable results between 1 and 20% (Figure R4). We tried sorting the transfected cells, as the MUC2 plasmids we received from the authors of the accompanying paper carry mCherry to select MUC2-positive cells, but even when combining 3 transfected wells (6well format) we could not sort enough cells for RNA extraction.

Figure R3. Low GFP plasmid transfection efficiency of LS174T cells using lipofectamine, assessed by flow cytometry. LS174T cells were transfected using an optimization range of Lipofectamine 3000 (left) and Lipofectamine LTX (right). Top panel shows untransfected cells, bottom panel shows the highest transfection rate obtained for each Lipofectamine reagent.

Figure R4. Variable nucleofection efficiency of LS174T cells, assessed by flow cytometry. LS174T cells were nucleofected with the 4 truncated MUC2 plasmids (see also accompanying paper by Neidhardt et al).

- As an alternative approach for expressing exogenous MUC2, we tried to boost endogenous MUC2 expression levels using various treatments found in literature. An experiment comparing 48-hour treatment using IL-13 (a type 2 cytokine, Iwashita et al.

Immunol Cell Biol. 2003), MSAB (a β -catenin inhibitor, which leads to LS174T cell differentiation into goblet cells (Van de Wetering et al. *EMBO Rep.* 2003) and butyrate (microbial metabolite known to induce goblet cells, Hatayama et al. *Biochem Biophys Res Commun.* 2007) however yielded no clear upregulation of MUC2 transcript and no clear IRE1 β -mediated endonuclease activity in our hands (Figure R5).

Figure R5. Assessment of goblet cell phenotype and XBP1 splicing after treatment using indicated compounds. LS174T^{ERN1-/-} cells were treated with the indicated compounds or combinations of compounds for 48 hours, after which gene expression was assessed via RT-qPCR.

- Because of the difficulties to study the role of the AGR2/IRE1 β axis on MUC2 regulation in LS174T cells, we had already started to establish the necessary mouse models to study AGR2-mediated regulation of IRE1 β in an endogenous setting. To this end, we have generated (and backcrossed) AGR2 and IRE1 β single deficient mice, compound deficient mice for IRE1 β and AGR2, and we are now crossing all these genotypes to MUC2-Cre reporter mice to be able to sort and analyze intestinal goblet cells. The mouse data look highly promising as the genetic ablation of AGR2 in intestinal epithelial cells does induce a strong decrease in Muc2 mRNA expression, which can be restored by compound deficiency of IRE1 β , pointing towards a potential role for IRE1 β RIDD in regulating Muc2 mRNA levels (Fig. R6).

*To address activation of IRE1 β in conditions of high mucus folding load (rather than deficiency of AGR2), we are optimizing models to boost endogenous mucin production, including *Trichuris muris* infection or using IL13 transgenic mice. This is all ongoing work and will require at least another 6 months to 1 year to get all the necessary crossings and establish the models.*

*So, while we do agree with the reviewer that this is a highly interesting and highly relevant question, we know the limitations of the LS174T system and would prefer to address this in an *in vivo* setting to maximize physiological relevance. Hence, this will be part of a follow-up story.*

Minor point

Figure 4E, the band migrating below IRE1 in the input should be labeled as non-specific if this is the case.

This has now been correctly labeled.

Referee #2:

This manuscript by Cloots et al employs a cell biological approach to identify AGR2 as a goblet-cell specific protein involved in the selective regulation of IRE1beta, but not IRE1alpha. Using mammalian cell culture models that express IRE1beta at either high (goblet cell derived L174T cells) or low levels (Calu cells), they depleted endogenous IRE1alpha and then expressed dox-inducible IRE1beta. They found that cells lacking endogenous IRE1beta showed higher basal levels of toxicity and IRE1 activity in response to dox-induced expression of IRE1beta, as compared to cells expressing endogenous IRE1beta. This suggested that cells that endogenously express IRE1beta (e.g., goblet cells) must also express some other factor that selectively represses IRE1beta activity. Using a proteomic-based approach to study interactomes of IRE1alpha or IRE1beta, they identified AGR2 as a specific protein that co-purifies with IRE1beta goblet-derived LS174T cells. They go on to show that overexpression of AGR2 can suppress toxicity and IRE1beta activity in

IRE1alpha-depleted Calu cells. They go on to demonstrate that AGR2 appears to disrupt IRE1beta dimers, providing a potential mechanism to explain the repression of IRE1beta activity observed for this protein. Finally, the authors show that catalytically inactive AGR2 (where a Cys in the thioredoxin active site is disrupted) and a disease-associated mutation (H117Y) impaired the ability to disrupt IRE1beta dimers and mitigate toxicity in Calu cells expressing IRE1beta. Collectively, these results identify AGR2 as a specific repressor of IRE1beta activity in specialized cells such as goblet cells.

This cell biological manuscript, in combination with the co-submitted manuscript from Neidhardt et al that applies a more biochemical approach, clearly demonstrates an important role for AGR2 in suppressing IRE1beta activity. These two manuscripts, which approach the problems in complementary ways, provide new mechanistic insight into this problem and highlight unique relationships between cell-type specific regulation of UPR stress sensors. The experiments in this manuscript are well performed and interpreted, all of which support the conclusions highlighted in the text. I do have a few minor comments that could be addressed in revision, but these two manuscripts, especially when viewed together, are quite strong in demonstrating an important role for AGR2 in regulating IRE1beta in specific cell types such as goblet cells.

COMMENTS.

1. The RIDD results in Figure 3A. I think it would be useful to repeat this a few more times to get a better idea on how AGR2 regulates RIDD in the Calu cells. The spread on the BLOC1S1 is quite large. I get the point that this reflects RNase activity and the XBP1s/t data is there too, but for completeness improving that figure a bit would be helpful.

In the meantime, this experiment has been conducted a total of 5 times, all these experiments will be included in the data source file. BLOC1S1 values generally show high variability, and 2 out of 5 experiments show ineffective inhibition of the endonuclease activity by 4u8C (likely due to the instability of the compound). However, as you can appreciate from Figure R8 below, it is consistent that the reduction of BLOC1S1 expression levels is always more pronounced in cells where only IRE1β is expressed compared to cells where both IRE1β and AGR2 are being co-expressed (red arrows, Fig R8), though AGR2 co-expression never blocks RIDD completely. The final experiment we conducted displayed the least variation between replicates, so this figure is now included as figure 3B. The interpretation in the text has also been attenuated as RIDD is clearly affected, even though not completely abolished.

Figure R8. Overview of all repeat experiment assessing *BLOC1S1* expression in untreated cells, after doxycycline-mediated induction of *IRE1 β* expression, and after doxycycline combined with 4u8C to inhibit *IRE1 β* endonuclease activity. These conditions were assayed in cells either with or without additional expression of *AGR2*.

2. The authors show that catalytically inactive and disease-associated *AGR2* mutants impair the ability for *AGR2* to inhibit *IRE1 β* dimerization and *IRE1 β* -associated cellular toxicity. It would be good to also include evidence for impaired *IRE1 β* RNAse repression in these experiments (monitoring *RIDD* targets and *XBP1s*).

This is a very relevant question and data supporting the reduced ability of H117Y and C81S to block IRE1 β endonuclease activity -both its ability to splice XBP1, as its ability to reduce Bloc1s1 levels- are now included in Figure 5, panels G (XBP1S) and H (BLOC1S1).

3. Could overexpression of mucin 'de-repress' the reduced *IRE1 β* activity observed in Calu cells expressing both *AGR2* and dox-inducible *IRE1 β* ? I realize that a similar experiment was performed in the accompanying paper, but it would be good to show similar results in this manuscript as well to provide some evidence to the idea that *AGR2* specifically senses mucin ER protein homeostasis. This is alluded to in the text and adding an experiment to address that would be nice to help paint the full picture in this specific manuscript.

A similar point was raised by reviewer #1, the answer to reviewer #1 is copied below:

This is a highly relevant question, but unfortunately highly challenging to address in our cell models for the following reasons:

- The Invitrogen ViraPower TRex dox-inducible system already takes up around 5kb of the viral genome, meaning only smaller genes of interest can be incorporated. We have already found how challenging it is to prepare virus carrying IRE1 α or IRE1 β using this system (which are under 3kb in size), and the largest MUC2 fragment tested in the accompanying paper is ~4.2kb in size. Obtaining sufficient titers and a validated inducible system was not feasible in the normal timeframe of a revision.

- To investigate this point in a shorter timeframe, we tested transient transfection, but LS174T cells turn out to be incredibly difficult to transfect. Lipofectamine 3000 and LTX yield transfection rates of maximum 3-5% (Figure R3), and Lonza nucleofection gave very variable results between 1 and 20% (Figure R4). We tried sorting the transfected cells, as the MUC2 plasmids we received from the authors of the accompanying paper carry mCherry to select MUC2-positive cells, but even when combining 3 transfected wells (6well format) we could not sort enough cells for RNA extraction.

Figure R3. Low GFP plasmid transfection efficiency of LS174T cells using lipofectamine, assessed by flow cytometry. LS174T cells were transfected using an optimization range of Lipofectamine 3000 (left) and Lipofectamine LTX (right). Top panel shows untransfected cells, bottom panel shows the highest transfection rate obtained for each Lipofectamine reagent.

Figure R4. Variable nucleofection efficiency of LS174T cells, assessed by flow cytometry. LS174T cells were nucleofected with the 4 truncated MUC2 plasmids (see also accompanying paper by Neidhardt et al).

- As an alternative approach for expressing exogenous MUC2, we tried to boost endogenous MUC2 expression levels using various treatments found in literature. An experiment comparing 48-hour treatment using IL-13 (a type 2 cytokine, Iwashita et al. Immunol Cell Biol. 2003), MSAB (a b-catenin inhibitor, which leads to LS174T cell differentiation into goblet cells (Van de Wetering et al. EMBO Rep. 2003) and butyrate (microbial metabolite known to induce goblet cells, Hatayama et al. Biochem Biophys Res Commun. 2007) however yielded no clear upregulation of MUC2 transcript and no clear IRE1 β -mediated endonuclease activity in our hands (Figure R5).

Figure R5. Assessment of goblet cell phenotype and XBP1 splicing after treatment using indicated compounds. LS174T^{ERN1-/-} cells were treated with the indicated compounds or combinations of compounds for 48 hours, after which gene expression was assessed via RT-qPCR.

- Because of the difficulties to study the role of the AGR2/IRE1 β axis on MUC2 regulation in cells, we had already started to establish the necessary mouse models to study AGR2-mediated regulation of IRE1 β in an endogenous setting. To this end, we have generated (and backcrossed) AGR2 and IRE1 β single deficient mice, compound deficient mice for IRE1 β and AGR2, and we are now crossing all these genotypes to MUC2-Cre reporter mice to be able to sort and analyze intestinal goblet cells. The mouse data look highly promising as the genetic ablation of AGR2 in intestinal epithelial cells does induce a strong decrease in Muc2 mRNA expression, which can be restored by compound deficiency of IRE1 β , pointing towards a potential role for IRE1 β RIDD in regulating Muc2 mRNA levels (Fig. R6).

*To address activation of IRE1 β in conditions of high mucus folding load (rather than deficiency of AGR2), we are optimizing models to boost endogenous mucin production, including *Trichuris muris* infection or using IL13 transgenic mice. This is all ongoing work and will require at least another 6 months to 1 year to get all the necessary crossings and establish the models.*

So, while we do agree with the reviewer that this is a highly interesting and highly relevant question, we know the limitations of the LS174T system and would prefer to address this in an in vivo setting to maximize physiological relevance. Hence, this will be part of a follow-up story.

Referee #3:

Accumulating misfolded proteins in the endoplasmic reticulum (ER) of eukaryotic cells trigger the so-called unfolded protein response (UPR), a transcriptional and translational response that activates the expression of molecular chaperones to cope with the misfolded proteins in the ER and down regulates general translation to reduce the folding load of incoming newly synthesized proteins. One of the central regulators of the UPR is Ire1, a kinase and endonuclease that activates the transcription factor XBP-1. In mammalian cells there are two isoforms of Ire1, the ubiquitously expressed Ire1 α and Ire1 β , which is only expressed in mucus producing cells like the goblet cells of the intestine. Whereas the regulation of Ire1 α is intensively investigated, regulation of Ire1 β was still poorly studied. Ectopic expression of Ire1 β in many generally used cell lines was shown to lead to growth arrest.

The authors of this study investigated the expression, activation and attenuation of Ire1 β in a cell line derived from goblet cells, LS174T, and, for comparison, the lung epithelial cell line Calu-1 that normally do not express Ire1 β . They confirmed that high level expression of Ire1 β exerts toxicity in Calu-1 cells but not in LS174T cells. They show that expression of Ire1 β induces splicing of XBP-1 mRNA and endonucleolytic degradation of mRNAs and that inhibition of these activities prevents the toxicity effect of Ire1 β expression, suggesting that there is a repressor present in LS174T cells that is absent in other cell lines tested. The authors immunoprecipitated Ire1 β and for comparison Ire1 α from LS174T cells and analyze the co-precipitating proteins by quantitative mass spectrometry. Among the most significantly enriched proteins that specifically co-precipitate with Ire1 β but not Ire1 α was AGR2, a chaperone and protein disulfide isomerase that was already linked to diseases caused by deficient mucus production of goblet cells. The authors demonstrate that Ire1 β is dimeric or oligomeric in vivo and that AGR2 interacts with Ire1 β in vivo and in vitro. They provide evidence that AGR2's interaction leads to monomerization of Ire1 β in cells. Co-expression of AGR2 with Ire1 β in Calu-1 cells alleviates the toxicity effect of Ire1 β overexpression. Down-regulation of AGR2 by siRNA in LS174T cells derepresses Ire1 β activity. Amino acid replacements, including a patient mutation, that compromises the interaction of AGR2 with Ire1 β leads to de-repression of Ire1 β and increased XBP-1 splicing activity. This is a very interesting study that is complementary to the parallel submitted study by Ron and coworkers (Neidhardt et al). There are a few points the authors should address before publication.

Major criticism

I assume that the supplemental Figures have been named extended view figures in the text and figure legends.

This has been stylistically finalized in the revised manuscript

A large body of evidence in the manuscript resides on immune blot analysis (often called quantification of protein levels by the authors in the figure legends). However, there is not a single quantitative analysis of the immune blots shown and no statistical significance of differences has been established. This is essential and has to be performed. Although, differences seem obvious in some of the blots (e.g. fig. 1D), in others (e.g. fig. 5c, or XBP1s in fig. S2B) it is not so clear how reproducible such blots were and whether similar changes were also found in repeated experiments. Co-precipitations should be normalized to the amount of the bait protein. If less bait protein is pulled down, it is expected that less prey protein is detected in the immune blot.

The semi-quantitative analyses and normalizations have been added in the revised figures that require quantification for proper interpretation and to indicate reproducibility of the effect (e.g. Fig S2B lower XBP1S expression in AGR2-expressing cells, alongside an increase in FLAG signal, Fig 3I, XBP1S protein induction after AGR2 siRNA in LS174T cells; Fig. 4C, IRE1 β -FLAG signal in gel filtration fractions; and Fig 5C, impairment of H117Y and C81S to disrupt IRE1 β complexes).

Quantification graphs include the quantifications of the replicates of that specific experiment. For all experiments we have also included the replicate data images in the source file, so this can be consulted by readers.

Minor comments:

Fig. 3E: the authors show relative expression of AGR2 after siRNA mediated knock-down. However, neither out of the figure nor of the figure legend it becomes clear whether this panel shows qPCR data quantification of the mRNA or immunoblot quantification of the protein levels. This should be stated clearly.

Fig. 3F the authors should clearly state how the changes were determined.

Fig. 6: since this figure has only a single panel the panel letter "A" is unnecessary and confusing. The same applies to Fig. S3/Fig. EV3.

Fig. EV4: the figure numbering is not shown on the figure.

All these comments have been addressed editorially

Dear Sophie,

Thank you for addressing the reviewers' comments. As you will see, you have addressed all concerns satisfactorily; therefore, unless any unexpected issues arise, I will not seek any additional input from the reviewers. Before I can finally accept the manuscript, there are some remaining editorial points which need to be addressed. In this regard, would you please:

- acknowledge funding from Medical Research Council DTP and Gates Cambridge PhD programme in our online submission system,
- limit the number of authors in the reference section to 10 authors + et al.,
- rename the 'conflict of interest statement' to the 'disclosure and competing interests statement',
- remove the author credit section from the manuscript,
- label the final panel in figure 1 'F',
- fill the manuscript number in the author checklist and complete the Source Data checklist,
- include a statement that the same image was used in Fig. 1C and FigEV1B in the Fig. EV1 legend,
- make dataset PXD042756 fully public upon acceptance of the manuscript,
- define 'n' in the legends of figures 1a; 3f; EV2a, b; EV3
- define error bars in the legends of figures 1e-f; 3a-b, 3d, f, g, i; EV3
- note that n=2 in figures 3i and 5f, and remove error bars from these panel. If it helps the readers' understanding, you may indicate the range of measurements with a bar, stating in the legend that range of measurement is indicated,
- include a scale bar and its definition in the legend for figures 1c; 3c; 5e; EV1b,
- indicate what the asterisk represents in the legend of figure 1d; 4e, and
- rearrange the manuscript order so that the main and EV figure legends are placed after the references.

I look forward to receiving these changes. EMBO Press is an editorially independent publishing platform for the development of EMBO scientific publications.

Best wishes,

William

William Teale, PhD
Editor
The EMBO Journal
w.teale@embojournal.org

We realize that it is difficult to revise to a specific deadline. In the interest of protecting the conceptual advance provided by the work, we recommend a revision within 3 months (1st Feb 2024). Please discuss the revision progress ahead of this time with the editor if you require more time to complete the revisions. Use the link below to submit your revision:

Referee #1:

The authors addressed most of the comments raised by the reviewers. I agree that the accompanying manuscript is quite complementary. Overall, this is a significant discovery supported by two well-performed and complementary studies.

Referee #2:

The authors have addressed all of my concerns from the previous submission with revisions to the text and additional experiments. I am satisfied that this manuscript is suitable for publication in EMBO.

Referee #3:

The authors studied the molecular mechanism of the regulation of the unfolded protein response by the stress sensor Ire1 β in the intestinal goblet cell derived cell line LS174T where it is expressed naturally and in lung cell line Calu-1 where Ire1 β is normally only lowly expressed. They demonstrate that Ire1 β is oligomeric and constitutively active unless repressed and monomerized by AGR2. The results are convincing in particular in combination with the accompanying manuscript by Neidhardt and colleagues. This study will be of great interest for a wide audience across several research fields. The authors addressed all comments appropriately. There are only two minor comments.

Major comments:

Fig. 3A: The authors state in their revision that "the BLOC1S1 values generally show high variability". They performed the experiment 5 times with variable results but then only want to show the 5th experiment that most closely shows the result the authors wish to see. I do not think that this is appropriate. If there are no very good reasons for excluding data (maybe ineffective inhibition due to instability of 4 μ 8c), all data should be combined and shown in one figure and statistical significance of differences should be tested by ordinary one-way ANOVA with Sidak's multiple comparison test. If necessary, data could be normalized to the average of the untreated control. It is important that readers realize the high variability of the experiment which impacts on the strength of the conclusions drawn. Only showing the 5th experiment insinuates more accuracy as actually observed.

Fig. 4B legend: It is very odd to write "Representative of one independent experiment". "Representative chromatogram for the chromatography runs analyzed in panel c" might be better.

The authors addressed the minor editorial issues.

Dear Sophie,

There are one or two outstanding (but small) issues with your latest submission. Could you please:

- use lower case 'm' for the SI unit for length in figures (if you wish to state the values here) and their legends,
- remove error bars for Figure EV3; they still indicate SE for a sample size of two,
- please use 'non-specific' instead of 'aspecific' in order to differentiate from 'a specific', and
- please acknowledge grant number MC_U105184326 in the manuscript text.

Best wishes,

William

William Teale, PhD
Editor
The EMBO Journal
w.teale@embojournal.org

We realize that it is difficult to revise to a specific deadline. In the interest of protecting the conceptual advance provided by the work, we recommend a revision within 3 months (8th Feb 2024). Please discuss the revision progress ahead of this time with the editor if you require more time to complete the revisions. Use the link below to submit your revision:

The authors addressed the remaining editorial issues.

Dear Sophie,

I am pleased to inform you that your manuscript has been accepted for publication in the EMBO Journal.

Congratulations - I really enjoyed working on this manuscript!

Best wishes,

William

William Teale, PhD
Editor
The EMBO Journal
w.teale@embojournal.org
